# DNA damage sensitivity of SWI/SNF-deficient cells depends on TFIIH subunit p62/GTF2H1

Cristina Ribeiro-Silva [1], Özge Z. Aydin[1,3], Raquel Mesquita-Ribeiro[2], Jana Slyskova[1], Angela Helfricht[1], Jurgen A. Marteijn[1], Jan H. J. Hoeijmakers[1], Hannes Lans [1] & Wim Vermeulen[1]

Mutations in SWI/SNF genes are amongst the most common across all human cancers, but efficient therapeutic approaches that exploit vulnerabilities caused by SWI/SNF mutations are currently lacking. Here, we show that the SWI/SNF ATPases BRM/SMARCA2 and BRG1/SMARCA4 promote the expression of *p62/GTF2H1*, a core subunit of the transcription factor IIH (TFIIH) complex. Inactivation of either ATPase subunit downregulates GTF2H1 and therefore compromises TFIIH stability and function in transcription and nucleotide excision repair (NER). We also demonstrate that cells with permanent BRM or BRG1 depletion have the ability to restore *GTF2H1* expression. As a consequence, the sensitivity of SWI/SNF-deficient cells to DNA damage induced by UV irradiation and cisplatin treatment depends on GTF2H1 levels. Together, our results expose GTF2H1 as a potential novel predictive marker of platinum drug sensitivity in SWI/SNF-deficient cancer cells.

[1] Department of Molecular Genetics, Oncode Institute, Erasmus MC, University Medical Center Rotterdam, Dr. Molewaterplein 40, 3015 GD Rotterdam, The Netherlands. [2] School of Life Sciences, University of Nottingham, NG7 2UH Nottingham, United Kingdom. [3] Present address: Molecular Biology and Genetics Department, Koç University, Istanbul 34450, Turkey. Correspondence and requests for materials should be addressed to H.L. (email: w.lans@erasmusmc.nl) or to W.V. (email: w.vermeulen@erasmusmc.nl)

Compiled sequencing efforts have revealed the high prevalence of mutations in chromatin remodeling genes across many different types of cancer[1,2]. Inactivating mutations in subunits of the SWI/SNF ATP-dependent chromatin remodeling complexes are amongst the most frequently mutated genes in human cancers[3,4], which argues for a major role in cancer pathogenesis. SWI/SNF complexes contain one of two mutually exclusive catalytic ATPase subunits, BRM/SMARCA2 or BRG1/SMARCA4, and multiple core and accessory subunits that together form a variety of functionally distinct complexes[5]. BRM and BRG1 use the energy of ATP to remodel chromatin, through which they regulate transcription, DNA damage repair (DDR) and replication and impact a variety of cellular processes including cell differentiation and growth[1,5,6].

Mutations in SWI/SNF subunits result in aberrant chromatin structures, increased genomic instability and perturbation of transcriptional programs, which are all hallmarks of cancer that can contribute to cell transformation and tumorigenesis[1,5–7]. Because the products of these typically loss-of-function mutations do not constitute obvious drug targets, efficient therapeutic strategies to target tumor cells with mutant SWI/SNF genes are still lacking. Detailed insight into the molecular mechanisms of the many anti-tumorigenic cellular functions of SWI/SNF is required in order to develop such strategies.

SWI/SNF proteins have been implicated in multiple DDR mechanisms, including double strand break (DSB) repair and nucleotide excision repair (NER), and are thought to coordinate signaling and efficient recruitment of repair proteins to chromatin[6,8,9]. NER removes a wide range of structurally unrelated helix-distorting DNA lesions, including cyclobutane pyrimidine dimers (CPDs) and 6–4 photoproducts (6–4PPs) induced by UV-light, ROS-induced cyclopurines and intrastrand crosslinks generated by chemotherapeutic platinum drugs[10,11]. If not repaired, these lesions interfere with transcription and replication, which can result in cell death or lead to mutations and genome instability that contribute to oncogenesis. Depending on the location of DNA lesions, two distinct DNA damage detection mechanisms can trigger NER. Transcription-coupled NER (TC-NER) is initiated when RNA Polymerase II is stalled by lesions in the transcribed strand and requires the CSB/ERCC6, CSA/ERCC8, and UVSSA proteins[11,12]. Global-genome NER (GG-NER) detects lesions anywhere in the genome by the concerted action of the damage sensor protein complexes UV-DDB, comprised of DDB1 and DDB2, and XPC-RAD23B-CETN2[13]. XPC and CSB are essential for the subsequent recruitment of the core NER factors to damaged DNA, starting with the transcription factor IIH (TFIIH)[12,14], a 10-subunit complex involved in both transcription initiation and NER[15]. In NER, the XPB/ERCC3 ATPase and the structural component p62/GTF2H1 of the TFIIH complex are thought to anchor the complex to chromatin, via an interaction with XPC[14,16,17], while the XPD/ERCC2 helicase is believed to unwind DNA and verify the presence of proper NER substrates[18]. Subsequent recruitment of XPA and RPA stimulates damage verification and facilitates the recruitment and correct positioning of the endonucleases XPF/ERCC4-ERCC1 and XPG/ERCC5, which excise the damaged strand[19]. After excision, the resulting single-stranded 22–30 nucleotide DNA gap is restored by DNA synthesis and ligation[11].

In vitro, NER is more efficient on naked DNA templates than on chromatinized DNA[20], on which it was found to be stimulated by yeast SWI/SNF[21], suggesting that chromatin remodeling is necessary to facilitate access to damaged DNA and efficient repair of lesions[8,9,20]. Using SWI/SNF mutant *C. elegans*, we found that SWI/SNF proteins protect organisms against UV irradiation, implying a role for SWI/SNF in promoting NER in vivo as well[22]. Several additional studies in yeast and mammals further indicate that SWI/SNF proteins are important for the UV-induced DDR[23–27]. However, conflicting observations on whether SWI/SNF regulates damage detection or facilitates later repair steps have made it difficult to deduce the exact mechanism underlying SWI/SNF activity in NER. Furthermore, the majority of studies have focused on the role of the BRG1 ATPase or the SNF5 subunit, but a putative role for BRM has never been investigated in detail.

In this study, we show that both BRM and BRG1 are necessary for efficient NER by promoting the expression of TFIIH subunit GTF2H1. Furthermore, we find that cells with permanent BRM or BRG1 loss have the ability to restore GTF2H1 levels. As a consequence, DNA damage sensitivity of BRM- or BRG1-deficient cells correlates with GTF2H1 protein levels, which could, potentially, be used to select SWI/SNF-deficient cancers that are more sensitive to platinum drug chemotherapy.

## Results

**SWI/SNF is required for efficient NER.** To test for SWI/SNF involvement in GG-NER, we measured UV-induced unscheduled DNA synthesis (UDS) in C5RO primary fibroblasts depleted of BRM or BRG1 by siRNA. BRM and BRG1 knockdown cells showed a clear decrease in UDS, comparable to cells in which the core NER factor XPA was depleted (Fig. 1a, b; Supplementary Fig. 1a). In addition, we measured recovery of RNA synthesis (RRS) after UV-C irradiation in U2OS cells depleted of SWI/SNF, to test involvement in TC-NER. After irradiation, transcription levels in cells with BRM or BRG1 knockdown failed to recover to the same degree as in control cells (Fig. 1c, d; Supplementary Fig. 1b). These results indicate that both BRM and BRG1 are essential for a robust GG- and TC-NER activity after UV irradiation.

To date, most efforts to study SWI/SNF function in NER have focused on BRG1, which prompted us to direct our efforts to BRM and to determine in which NER step this SWI/SNF ATPase plays a role. We used immunofluorescence (IF) to monitor the recruitment of endogenous key NER proteins to local UV-C damage (LUD)—induced by irradiation through a microporous membrane-, 30 min after damage induction in siBRM treated U2OS cells. Recruitment of the early DNA damage sensors DDB2 and XPC to LUD, marked by CPD staining, was unaffected by BRM depletion (Fig. 1e, f, Supplementary Fig. 1c). We validated these results by real-time confocal imaging of XPC-GFP recruitment to LUD induced by a 266 nm microbeam laser, which confirmed that XPC assembly kinetics were unchanged after BRM depletion (Supplementary Fig. 1d). Also, recruitment of CSB, which is difficult to assess using IF, to microbeam LUD was unaffected by BRM depletion (Supplementary Fig. 1e). Strikingly, however, BRM depletion significantly reduced the recruitment to LUD of the TFIIH proteins XPB, XPD, and GTF2H1 and downstream proteins XPA and XPF, as measured by IF (Fig. 1e, f). These results show that BRM does not facilitate lesion detection in GG- and TC-NER, but is required for the recruitment of the downstream NER damage verification and excision machinery, thus explaining why NER is compromised in its absence.

**BRM is required for the recruitment of TFIIH to chromatin.** To dissect how BRM depletion impairs NER, we focused on the TFIIH complex and measured real-time XPB-GFP accumulation at 266 nm laser-induced LUD, which was significantly lower (more than twofold) after BRM knockdown (Fig. 2a, b, Supplementary Fig. 1f). We confirmed this result with an additional independent siRNA (siBRM#2) to exclude siRNA off-target effects (Supplementary Fig. 1g). Using fluorescence recovery

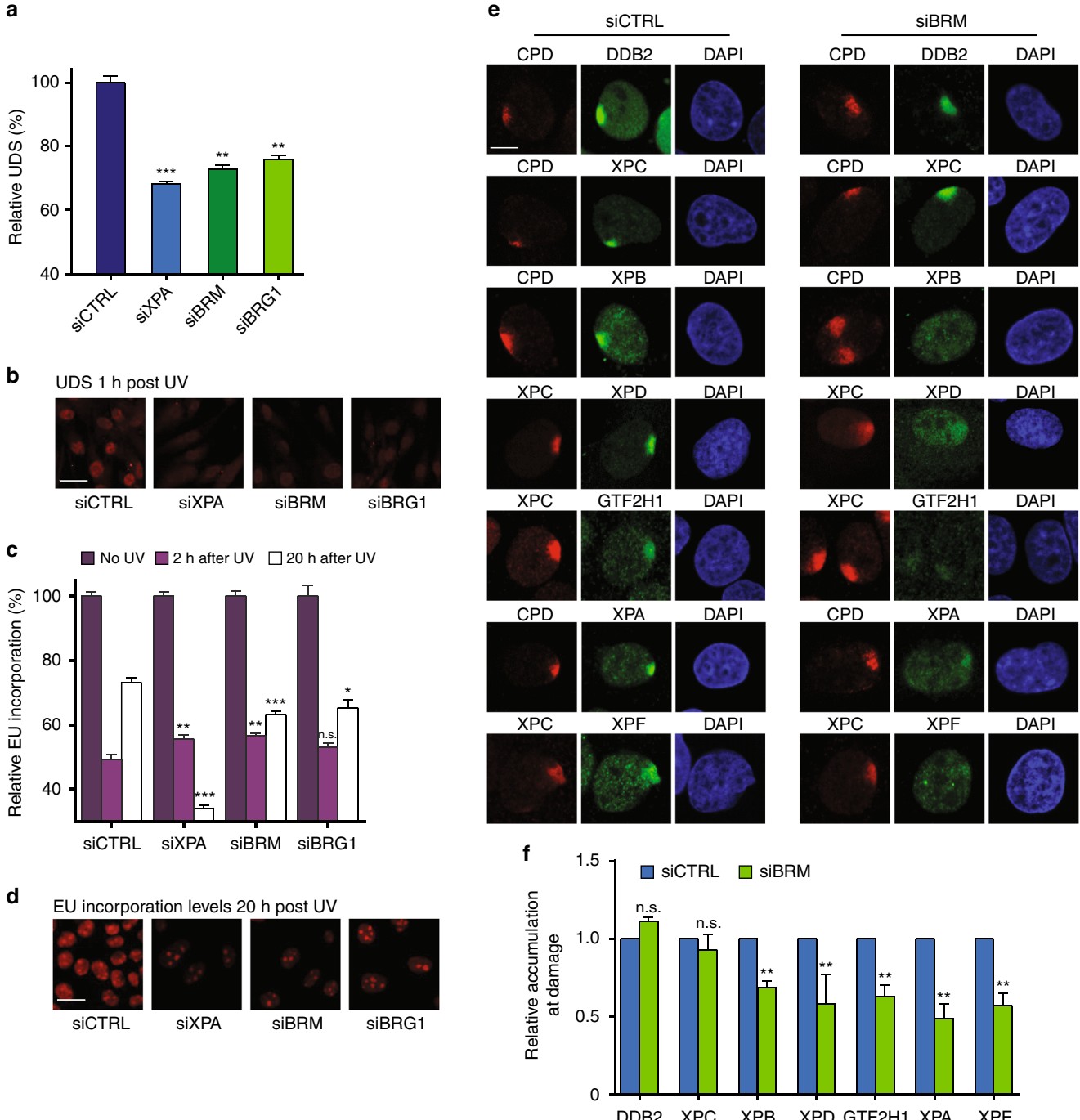

**Fig. 1** SWI/SNF is required for efficient NER. **a** Quantification of unscheduled DNA synthesis (UDS) in C5RO primary fibroblasts treated with non-targeting control (CTRL), XPA, BRM, and BRG1 siRNAs (Supplementary Fig. 1a). UDS was determined by EdU incorporation for 1 h after UV-C (16 J/m$^2$) irradiation followed by fluorescent staining of the incorporated EdU. Fluorescence was quantified and normalized to control, set to 100%. Mean & S.E.M. of > 200 cells per sample from two independent experiments. **P < 0.01, ***P < 0.001, relative to siCTRL. **b** UDS representative pictures, 1 h after UV-C. Scale bar: 25 μm. **c** Quantification of recovery of RNA synthesis (RRS) in U2OS cells treated with non-targeting control (CTRL), XPA, BRM, and BRG1 siRNAs (Supplementary Fig. 1b). Transcription levels in non-irradiated cells and in cells 2 and 20 h after UV-C irradiation (6 J/m$^2$) were determined by a 2 h pulse-labeling with the uridine analogue EU and subsequent fluorescent staining and measurement of incorporated EU. RRS levels were normalized to non-irradiated cells, set to 100%. Mean and S.E.M. of > 200 cells per condition from at least two independent experiments. *P < 0.05, ***P < 0.001, relative to each siCTRL in each time point. **d** RRS representative pictures, 20 h after UV-C irradiation. Scale bar: 25 μm. **e** Immunofluorescence (IF) showing recruitment of the indicated NER proteins (green channel) to local UV-C damage (LUD) in U2OS cells treated with control or BRM siRNAs (Supplementary Fig. 1c). Cells were fixed 30 min after inducing LUD with UV-C irradiation (60 J/m$^2$) through a microporous membrane (8 μm). UV lesions were marked with staining against CPD or XPC, red channel. DNA was stained with DAPI. Scale bar: 5 μm. **f** Quantification of NER proteins recruitment to LUD. Relative accumulation at LUD (over nuclear background) after siBRM was normalized to control, in which nuclear background was set at 0 and maximal signal at LUD set to 1.0 for each protein. Mean and S.E.M. of > 100 cells per sample, of at least two independent experiments, except for GTF2H1 which was only performed once. **P < 0.01, relative to siCTRL, n.s. non-significant

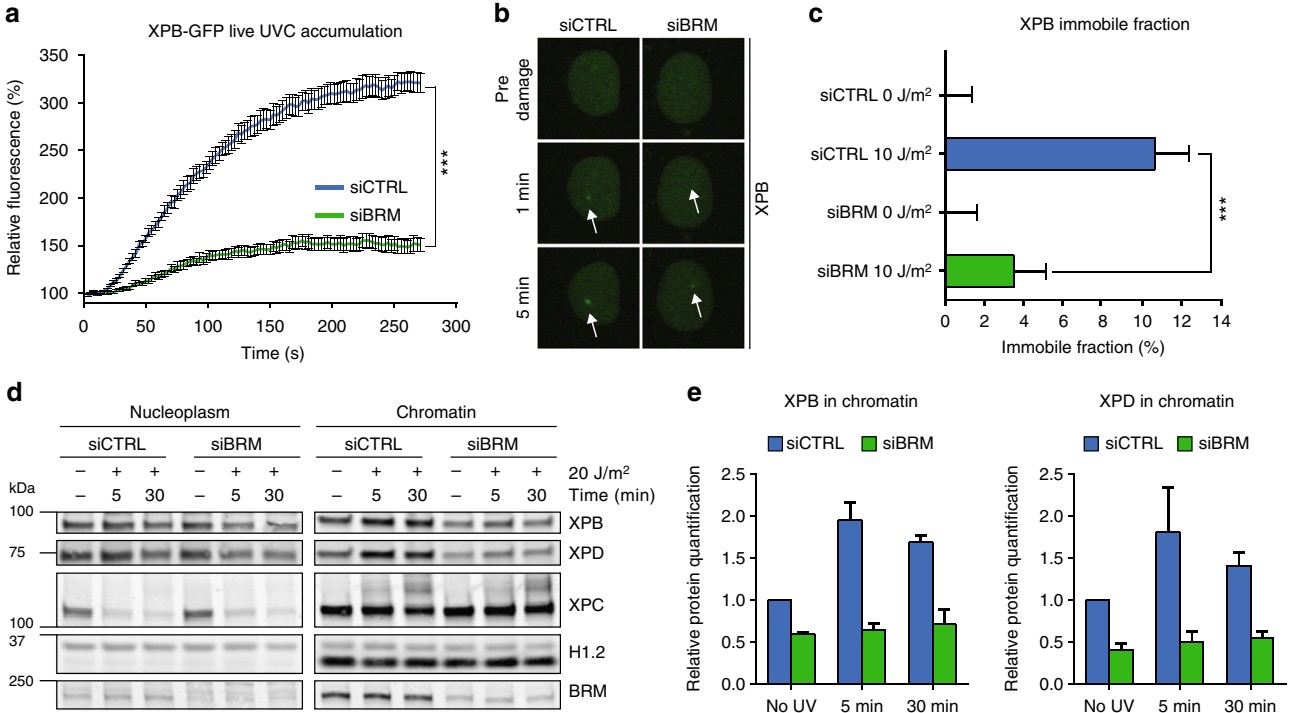

**Fig. 2** BRM is required for the recruitment of TFIIH to chromatin. **a** Real-time imaging of XPB-GFP accumulation at 266 nm UV-C laser-induced LUD in XPCS2BA cells treated with control and BRM siRNA (siCTRL and siBRM, respectively; Supplementary Fig. 1f). Pre-damage fluorescence intensity (nuclear background) was set to 100% ($t = 0$). Mean & S.E.M. of three independent experiments each with more than ten cells per condition. $P < 0.001$, compared to siCTRL. **b** Representative images of real-time recruitment of XPB-GFP, which resides exclusively in the nucleus, to laser generated LUD. Arrows indicate LUD regions. **c** Quantification of XPB-GFP immobile fraction in XPCS2BA fibroblasts. The mobility of XPB-GFP was determined by fluorescence recovery after photobleaching (FRAP) in mock and UV-C irradiated (10 J/m$^2$) cells treated with non-targeting control (CTRL) or BRM siRNAs, as depicted in Supplementary Fig. 1h. The UV-induced immobile fraction (mean & S.E.M. from three independent experiments, with at least ten cells measured per condition each time) was determined as described in Supplementary Fig. 1h. ***$P < 0.001$ relative to UV-irradiated siCTRL. **d** Immunostaining of soluble (nucleoplasm) and chromatin-bound XPB, XPD, XPC, BRM, and H1.2 (as loading control) in U2OS cells treated with non-targeting control (CTRL) or BRM siRNAs. Cells were collected for protein fractionation at different time points after UV-C irradiation (20 J/m$^2$). **e** Relative quantification of chromatin-bound XPB and XPD, normalized to non-irradiated siCTRL, set to 1.0. Mean & S.E.M. of two independent experiments. Full-size immunoblot scans are provided in Supplementary Fig. 6a

after photobleaching (FRAP), we also measured UV-induced XPB-GFP immobilization. As previously observed[28], a fraction of XPB immobilized in response to UV-C irradiation in control conditions, as a result of TFIIH binding to UV-damaged DNA (Supplementary Fig. 1h). However, this UV-induced XPB immobilization was substantially reduced when BRM was depleted by siRNA (Supplementary Fig. 1h and quantified in Fig. 2c). These results further corroborate our IF experiments (Fig. 1e, f) and suggest that BRM is needed for efficient damage loading of TFIIH.

We also assessed damage-induced chromatin loading of TFIIH in U2OS cells with cellular fractionation, which confirmed that UV-induced loading of TFIIH subunits XPB and XPD, but not of XPC, was strongly reduced after BRM depletion (Fig. 2d, e). Strikingly, even in the absence of DNA damage, TFIIH association with chromatin was reduced, whereas its non-chromatin bound pool did not change significantly after BRM knockdown (Supplementary Fig. 2a). This implies that TFIIH is unable to efficiently interact with DNA irrespective of whether there is DNA damage or not. In addition, we noticed that association of BRM itself with chromatin did not change after DNA damage (Fig. 2d). We also could not detect recruitment of BRM to LUD inflicted by irradiation through a microporous membrane on IF (Supplementary Fig. 2b) and did not observe recruitment of GFP-tagged BRM to LUD inflicted by 266 nm

microbeam laser, as analyzed by real-time confocal imaging (Supplementary Fig. 2c). These results suggest that BRM is not actively recruited to sites of UV damage. Moreover, immuno-precipitation of XPB-GFP did not reveal an interaction of TFIIH with BRM, neither in the presence nor absence of UV-DNA damage (Supplementary Fig. 2d), while GTF2H1 was successfully co-purified with XPB-GFP, as expected. These observations indicate that BRM is not associated with TFIIH nor directly involved in its recruitment to chromatin, but suggest that BRM affects TFIIH chromatin binding in another way, possibly by regulating its general activity, stability or expression of its subunits.

**BRM stabilizes TFIIH by promoting *GTF2H1* expression**. The TFIIH complex consists of ten subunits and becomes unstable if one of these is impaired[15,29–31]. Given the fact that SWI/SNF acts in transcription regulation, we considered the possibility that BRM transcriptionally regulates one or more TFIIH genes. Therefore, we analyzed the individual expression of all TFIIH genes by real-time-qPCR (RT-qPCR) in U2OS cells after BRM knockdown. While expression of most TFIIH genes was unaffected by BRM knockdown, *GTF2H1* expression was strongly reduced (Fig. 3a). Immunoblot analysis revealed that this also resulted in lowered GTF2H1 protein levels (Fig. 3b), which we

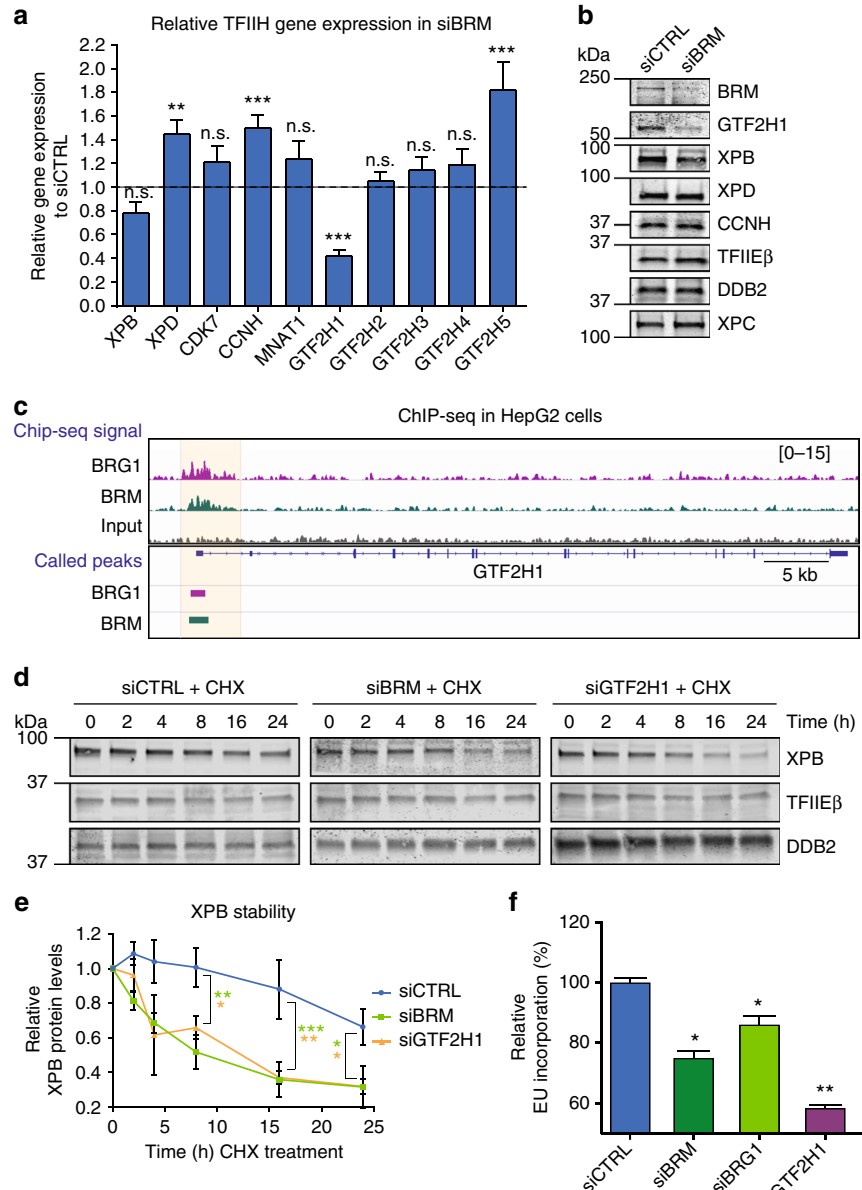

**Fig. 3** BRM stabilizes TFIIH by promoting GTF2H1 expression. **a** Relative quantification of individual TFIIH genes expression in U2OS cells treated with control (CTRL) or BRM siRNAs, as determined with RT-qPCR. Individual basal gene expression in BRM knockdown was normalized to siCTRL levels, which were set to 1.0 (dotted line in graph). *GAPDH* expression was used for normalization. Mean & S.E.M. of at least three independent experiments. **$P < 0.01$, ***$P < 0.001$ relative expression in each gene to siCTRL. n.s., non-significant. **b** Immunoblot analysis of TFIIH protein levels (GTF2H1, XPB, XPD, CCNH), TFIIEβ, DDB2 and XPC from whole cell extracts of U2OS treated with control (CTRL) or BRM siRNAs. Representative immunoblots of two independent experiments. **c** BRG1 and BRM co-occupancy of *GTF2H1* promotor. Re-analysis of published ChIP-seq data in which ChIP-seq signal density (top) and respective peaks (bottom) illustrate BRG1 (purple) and BRM (green) enrichment at the promoter of *GTF2H1* in HepG2 cells (upon shNS transfection[32]). Promoter region of interest highlighted in light orange, signal density in reads per million. **d** XPB protein stability was evaluated in U2OS cells treated with control (CTRL) or BRM siRNAs at different time points after addition of 100 μM cycloheximide (CHX) to inhibit protein synthesis. Immunostainings of TFIIEβ and DDB2 were used as negative and loading controls, respectively. **e** Quantification of XPB protein levels normalized to DDB2 in time after addition of CHX. The total amount of XPB in whole cell lysates was set to 1.0 at $t = 0$. Mean & S.E.M. of at least three independent experiments. *$P < 0.05$, **$P < 0.01$, ***$P < 0.001$ for each time point of siBRM (green) or siGTF2H1 (orange) relative to siCTRL. **f** Relative quantification of transcription levels in U2OS cells treated with non-targeting control (CTRL), BRM, BRG1, or GTF2H1 siRNAs. Transcription was determined by measuring EU incorporation in non-irradiated cells 48 h after siRNA treatment. EU relative fluorescence intensity was set to 100% in siCTRL treated cells. Mean & S.E.M. of > 200 cells from two (siGT2H1) and three (siBRM and siBRG1) independent experiments. Full-size immunoblot scans are provided in Supplementary Fig. 6b, c

further corroborated by IF staining of GTF2H1 after BRM depletion using an independent siRNA (siBRM#2), to exclude siRNA off-target effects (Supplementary Fig. 3a, b). Besides GTF2H1, we also found mildly reduced expression of *XPB*, both

at the mRNA and protein level. In contrast, protein levels of XPD and CCNH—whose mRNA levels were mildly increased, and of TFIIEβ, XPC, and DDB2 were unaltered after BRM depletion (Fig. 3a, b). To verify that BRM can regulate *GTF2H1*

transcriptionally, we re-analyzed published whole-genome BRM ChIP-seq data for HepG2[32] and RWPE1[33] cells. In both cell types we observed an enrichment of BRM ChIP-seq signal at the *GTF2H1* promoter region, suggesting the association of BRM with active regulatory regions of the *GTF2H1* gene (Fig. 3c, Supplementary Fig. 3c). These results therefore suggest that BRM promotes *GTF2H1* expression and may explain why BRM depletion leads to defects in TFIIH chromatin loading, as GTF2H1 was shown to be essential for the structural integrity of the TFIIH complex[31].

To assess whether TFIIH indeed becomes unstable in the absence of BRM, we determined the half-life of XPB in BRM-depleted U2OS cells after blocking protein synthesis with cycloheximide (CHX) treatment. Quantification of XPB protein levels, normalized to DDB2, revealed a strongly accelerated proteasome-dependent degradation of XPB in the absence of BRM (Fig. 3d, e; Supplementary Fig. 3d). Importantly, XPB was similarly less stable in cells depleted of GTF2H1 by siRNA (Fig. 3d, e). To confirm that BRM depletion specifically affected TFIIH and not other transcription factors as well (whose DNA-binding might be regulated by BRM[5,34]), we tested the stability of subunit beta of transcription initiation factor IIE (TFIIEβ). TFIIEβ is involved in recruiting TFIIH to the transcription initiation complex[35], but its stability was not affected by BRM knockdown (Fig. 3d, Supplementary Fig. 3e). These results, therefore, suggest that the TFIIH complex is less stable in the absence of BRM because of reduced amounts of GTF2H1 that limit the stable assembly of functional TFIIH complexes. This likely impairs the stability of TFIIH subunits and TFIIH function in transcription and NER. Indeed, either BRM or GTF2H1 depletion also reduced transcription levels in U2OS cells, likely due to limiting amounts of TFIIH (Fig. 3f).

**GTF2H1 expression rescues TFIIH function in BRM/BRG1 depleted cells**. To demonstrate that impaired TFIIH function in BRM knockdown cells is mainly a consequence of GTF2H1 downregulation, we tested if ectopic expression of GFP-GTF2H1 or XPB-GFP (as control) reversed impaired TFIIH DNA damage recruitment. Overexpression of both TFIIH subunits did not affect XPD recruitment to LUD in control U2OS cells (Fig. 4a, b). However, overexpression of GFP-GTF2H1, but not of XPB-GFP, rescued XPD accumulation to LUD in BRM and GTF2H1 depleted cells, confirming that reduced GTF2H1 expression, as a consequence of BRM depletion, impairs TFIIH function.

Since BRG1 depletion also resulted in GG- and TC-NER defects (Fig. 1a–d), similar to BRM, we tested whether BRG1 knockdown also affected TFIIH function via GTF2H1. Depletion of BRG1 led to lower overall transcription (Fig. 3f) and reduced GTF2H1 protein levels, as assessed by both immunoblot (Supplementary Fig. 3f) and IF using independent siRNAs to exclude off-target effects (Supplementary Fig. 3g, h). BRG1 was furthermore found to co-occupy the *GTF2H1* promoter together with BRM (Fig. 3c, Supplementary Fig. 3c). Also, BRG1 depletion led to reduced XPD recruitment to LUD (Supplementary Fig. 3i), which was rescued by ectopic expression of GTF2H1, but not of XPB (Fig. 4b, c). BRG1 did not localize to LUD induced by irradiation through a microporous membrane (Supplementary Fig. 2b) or by 266 nm microbeam laser (Supplementary Fig. 2c), implying that the protein itself does not directly participate in the NER reaction. Moreover, both siBRM and siBRG1 did not alter cell cycle distribution (Supplementary Fig. 3j) nor did they further decrease reduced XPD recruitment following GTF2H1 depletion (Supplementary Fig. 3k), indicating that BRM and BRG1 do not impair TFIIH recruitment due to indirect effects on the cell cycle or independently of GTF2H1. Overall, these results indicate that

the activity of both BRM and BRG1 is necessary to ensure normal GTF2H1 levels and TFIIH function, and, therefore, NER performance.

**Chronic BRG1-deficient cancer cells restore GTF2H1**. Because BRM and BRG1 are frequently mutated in cancer[3], we investigated if cancer cell lines with SWI/SNF mutations showed low GTF2H1 protein levels, as these cells would then likely be more susceptible to DNA damaging chemotherapeutic drugs. Unexpectedly, BRG1-deficient non-small cell lung cancer (NSCLC) lines A549 and H1299[36–38] showed normal GTF2H1 levels in comparison to U2OS (Fig. 5a, b). Strikingly, however, BRM knockdown in these NSCLC cell lines resulted in lower GTF2H1 expression, demonstrating that SWI/SNF-mediated expression of GTF2H1 is not cell type-specific. BRG1 knockdown only resulted in lower GTF2H1 levels in U2OS cells, which are wild-type for BRG1, but not in the BRG1-deficient A549 and H1299 cell lines (Fig. 5a, b), confirming again that GTF2H1 downregulation in U2OS cells is not due to an siRNA-mediated off-target effect. We next tested GTF2H1 protein levels by IF in additional BRG1 and/or BRM-deficient cancer cell lines. However, also BRG1-deficient Panc-1 and Hs 700T cells, BRM-deficient A2780 cells and BRM/BRG1-deficient SW13 and C33A cells, all consistently showed normal or even increased GTF2H1 levels, as compared to MRC5, Hs 578 T and U2OS cells (Supplementary Fig. 4a,b). The puzzling finding that chronic BRG1 and/or BRM deficiency in these cancer cell lines does not lead to permanent downregulation of GTF2H1, whereas transient depletion does, indicates that there might be an adaptive, compensatory mechanism in these cells that restores GTF2H1 expression to prevent chronic TFIIH dysfunction.

BRM and BRG1 have been shown to be able to compensate for some of each other's functions[36,39] and in many BRG1-deficient cancer cells including A549 and H1299, BRM has even become essential for cellular growth[36,38,40]. To test if regulation of GTF2H1 levels are in part responsible for BRM having become essential in BRG1-deficient cells, we generated A549 and H1299 cell lines stably expressing GFP-GTF2H1 (Fig. 5c, Supplementary Fig. 4c). siRNA-mediated BRM knockdown in these cells only reduced the expression of endogenous GTF2H1 (Fig. 5c, Supplementary Fig. 4c, d), guaranteeing that expression of the *GFP-GTF2H1* transgene, driven by the ectopic PGK promoter, is preserved even in the absence of both SWI/SNF ATPases. We then performed colony forming assays and found that siRNA-mediated depletion of BRM led to profound growth inhibition of BRG1-deficient A549 and H1299 cells. This, however, was not rescued by stable GFP-GTF2H1 expression (Fig. 5c–e, Supplementary Fig. 4e). As expected, control and BRG1 siRNA did not affect the proliferation capacity of these BRG1-deficient cells. These results indicate that synthetic lethality induced by BRM depletion in BRG1-deficient cancer cells is not dependent on GTF2H1 expression and likely involves other functions of these ATPases.

**DNA damage sensitivity of BRM cells correlates with GTF2H1 levels**. To confirm that cells can restore GTF2H1 expression as adaptation to chronic SWI/SNF dysfunction and to investigate the functional consequences, we permanently knocked out BRM and BRG1 in immortalized MRC5 fibroblasts, by transfection with sgRNAs targeting either *BRM* (sgBRM) or *BRG1* (sgBRG1). After careful selection of transfected cells, we confirmed by immunoblotting that this heterogeneous pool of transfected cells showed an overall highly efficient depletion of BRM or BRG1 and a concomitant downregulation of GTF2H1 levels (Fig. 6a, b). However, after culturing cells for multiple passages, IF of the heterogeneous pool of BRM and BRG1 knockout cells revealed

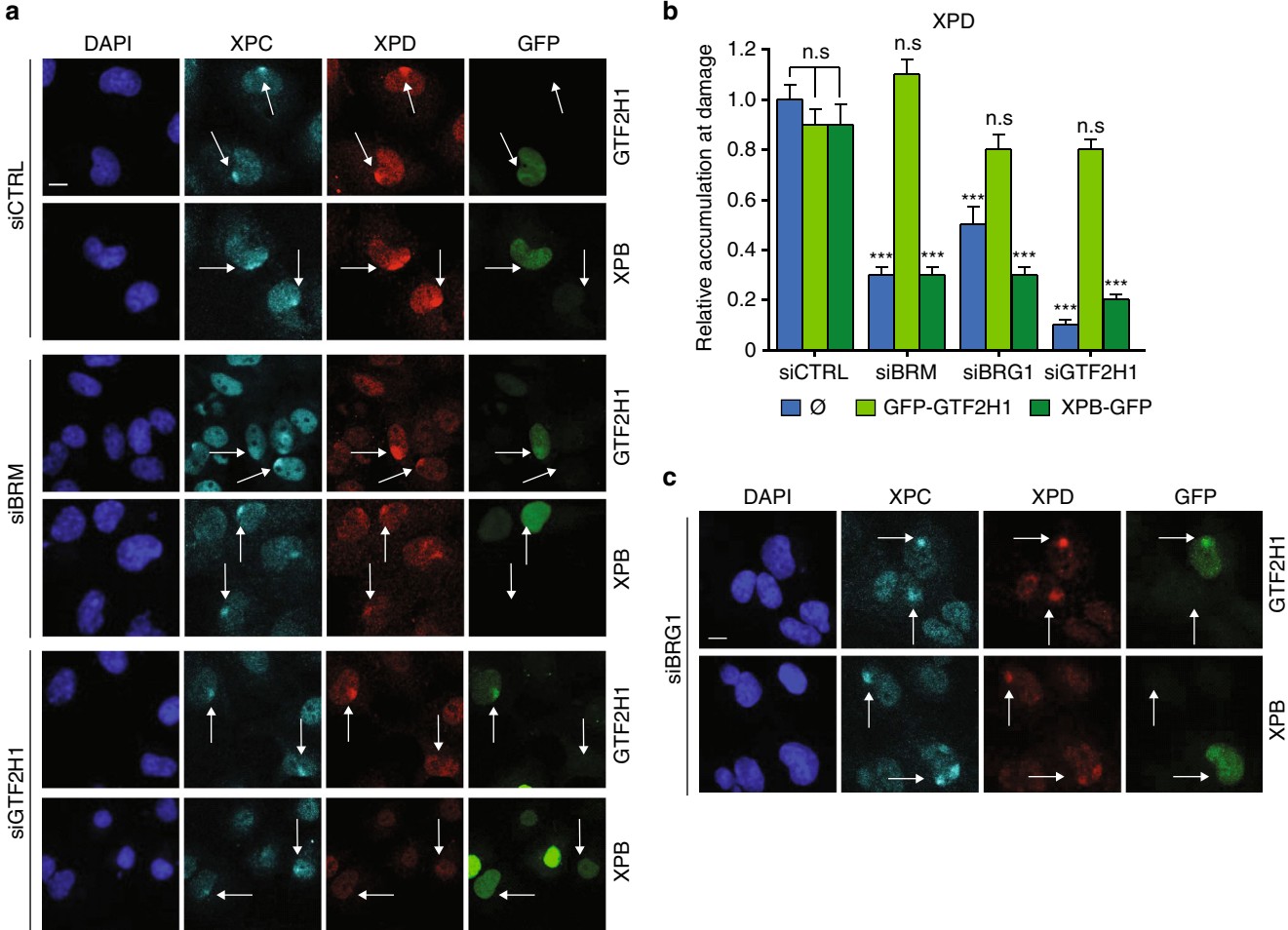

**Fig. 4** GTF2H1 expression rescues TFIIH in BRM/BRG1 depleted cells. Representative IF of XPD recruitment (red channel) to LUD marked by XPC (cyan channel). U2OS cells were fixed 30 min after local UV-C irradiation (60 J/m²) through a microporous membrane (8 μm). **a** U2OS cells were treated with control (CTRL), BRM, or GTF2H1 siRNAs and transiently transfected with TFIIH subunits XPB or GTF2H1 fused to GFP (green channel). Scale bar: 10 μm. **b** Quantification of XPD recruitment to LUD. Relative accumulation at LUD (over nuclear background) in each condition was normalized to control (siCTRL without transient transfection of TFIIH subunits, indicated by "empty" symbol), in which nuclear background was set at 0 and maximal signal at LUD set to 1.0 (>50 cells per sample, mean & S.E.M. from four independent experiments). ***$P < 0.001$, relative to siCTRL without transient transfection of TFIIH subunits. **c** U2OS cells were treated with siRNA against BRG1 and transiently transfected with TFIIH subunits XPB or GTF2H1 fused to GFP (green channel). Scale bar: 10 μm. Arrows highlight LUD in a mixed population of non-transfected and transfected cells with GFP-GTF2H1 or XPB-GFP (green cells in the right panel). n.s. non-significant

that individual cells had either retained the low GTF2H1 expression or restored it to wild-type level (Supplementary Fig. 5a). Establishment of multiple clonal cell lines from the MRC5 sgBRM pool of cells showed that many clones exhibited normal GTF2H1 levels, despite having no detectable BRM expression (Supplementary Fig. 5b). These striking findings show that cells are often able to adapt to the loss of one of the ATPase SWI/SNF subunits by restoring normal GTF2H1 expression levels.

We next selected two clones with low (c1 and c6) and two clones with normal (c3 and c7) GTF2H1 expression and confirmed the reduced and rescued GTF2H1 levels and BRM knockout by IF and immunoblot (Fig. 6c–f) and by sequencing the sgBRM target region (Supplementary Fig. 5c). Transient expression of BRM-GFP in c1 cells increased GTF2H1 expression (Fig. 6g), clearly demonstrating not only that the lower GTF2H1 levels are caused by BRM depletion but also that these are reversible. Transient BRG1-GFP expression, however, did not increase GTF2H1 protein levels in these cells (Fig. 6g). Likewise,

stable ectopic expression of GFP-tagged BRG1 in U2OS cells did not prevent the reduction in GTF2H1 levels after siBRM treatment (Supplementary Fig. 5d). These results suggest that restoration of GTF2H1 levels, as observed in cells with chronic BRM/BRG1 deficiency, is likely not due to compensation by the other ATPase.

Due to the central function of TFIIH in NER, we considered whether GTF2H1 levels in BRM knockout cells correlate with NER capacity and thus with sensitivity to DNA damaging agents. XPD recruitment to LUD was severely impaired in c1 cells with low GTF2H1 levels, but not in c3 cells with normal GTF2H1 levels (Fig. 6h, i). Clonal UV-survival assays corroborated these observations, showing that only c1 cells were UV-hypersensitive (Fig. 6j). These intriguing results could imply that cancer cells that have lost the activity of SWI/SNF subunit(s) may be differentially sensitive to DNA damaging chemotherapeutics depending on their GTF2H1 levels. Platinum-based drugs such as cisplatin are widely administered to treat various types of solid tumors[41] and kill cells by inducing DNA intra- and interstrand

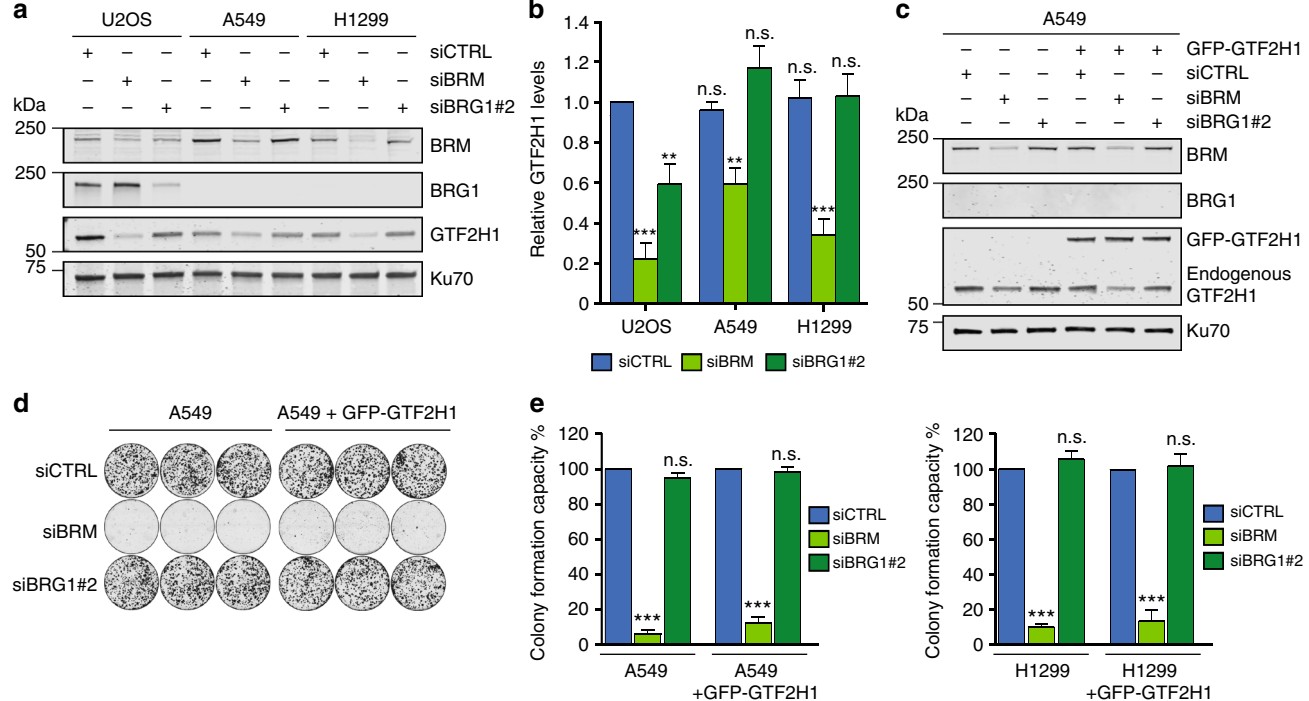

**Fig. 5** Cancer cells with chronic BRG1 deficiency restore GTF2H1 expression. **a** Immunoblot showing total protein levels of BRM, BRG1, and GTF2H1, in cell lysates of U2OS and BRG1-deficient non-small lung cancer cell (NSCLC) lines A549 and H1299 treated with control (CTRL), BRG1 or BRM siRNAs. Ku70 was used as loading control. **b** Relative quantification of GTF2H1 protein levels in U2OS, A549, and H1299 cells transfected with control (CTRL), BRG1 or BRM siRNA. GTF2H1 levels were normalized to Ku70 and the total relative amount of GTF2H1 in whole cell lysates was set to 1.0 in U2OS siCTRL. Mean & S.E.M. from at least three independent experiments **$P < 0.01$, ***$P < 0.001$, n.s., non-significant. **c** A549 cells with and without stable expression of GFP-GTF2H1, driven by the ectopic PGK promoter, were treated with control (CTRL), BRM or BRG1 siRNAs. Cell lysates were analyzed by immunoblotting against BRM and GTF2H1. Ku70 was used as loading control. **d** A549 cells, with or without stable expression of GFP-GTF2H1 were seeded 48 h after transfection with control (CTRL), BRM or BRG1 siRNAs, in triplicate, at a density of 1000 cells per well and grown for 12 before fixation and staining. **e** Quantification of colony forming capacity of A549 (shown in **d**) and H1299 (shown in Supplementary Fig. 4e) cell lines with or without stable GFP-GTF2H1 expression and treated with control (CTRL), BRM or BRG1 siRNAs. Clonal capacity was normalized to 100% in control conditions (CTRL). Mean & S.E.M. of three independent experiments, each performed in triplicate. ***$P < 0.001$, n.s., non-significant, relative to siCTRL. Full-size immunoblot scans are provided in Supplementary Fig. 7a,b

crosslinks that are mainly repaired by NER[42] and interstrand crosslink repair. Therefore, we tested cisplatin sensitivity of c1 and c3 cells to evaluate if this also correlates with their GTF2H1 expression levels. Markedly, c1 cells, but not c3 cells, showed increased sensitivity to cisplatin (Fig. 6k). To verify these findings, we also tested DNA damage sensitivity of BRM knockout clones c6 and c7, exhibiting respectively low and restored GTF2H1 levels (Fig. 6c–f). UV and cisplatin survival of these clones (Fig. 6l, m) confirmed that indeed GTF2H1 levels in BRM knockout cells determine NER capacity and sensitivity to DNA damage. These results indicate that loss of BRM sensitizes cells to cisplatin only if GTF2H1 protein levels are lowered, and imply that GTF2H1 levels could be used as a predictive marker for platinum drug sensitivity of SWI/SNF-deficient cancers.

## Discussion
Inactivating mutations in SWI/SNF subunit genes are amongst the most recurrent mutations found in all human cancers[3,4]. For instance, *BRG1* is mutated in 90% of small cell ovarian, 27% of skin and 5% of small cell lung cancers[1,7,37]. The homologous SWI/SNF ATPase *BRM* is also recurrently lost in multiple primary tumors and cancer cell lines, such as in over 15% of lung, ovarian and breast cancers[43] and was found to protect mice against UV-induced skin cancer[44]. It is thus advantageous to identify general vulnerabilities caused by SWI/SNF deficiency in

pathways with anti-tumorigenic function, to create opportunities for the development of effective therapeutic approaches.

In this study, we show that both BRM and BRG1 promote normal TFIIH function in transcription and NER by regulating the expression of the *GTF2H1* gene (Fig. 7). Both RT-qPCR and immunoblot analysis revealed significantly lower expression of GTF2H1 and mildly lower expression of XPB after BRM knockdown. Both these TFIIH subunits are required for recruitment of the TFIIH complex to damaged DNA[14,16], but only the ectopic expression of GTF2H1—not of XPB—rescued the binding of TFIIH to DNA damage in BRM and BRG1 depleted cells. This shows that lowered levels of GTF2H1, caused by BRM or BRG1 knockdown, act as a limiting factor for the assembly of functional TFIIH complexes, in agreement with recent literature describing GTF2H1 as essential for TFIIH complex integrity and stability[31,45]. Limiting amounts of functional TFIIH complexes likely impair overall TFIIH functions, in accordance with the observed decreased transcription levels and lower NER performance (Fig. 7).

ATP-dependent chromatin remodelers like SWI/SNF are thought to make chromatin more accessible to DNA repair proteins[8,9,20]. In line with this idea, the yeast Snf5 and Snf6 SWI/SNF subunits were shown to bind XPC and mediate UV-induced nucleosome remodeling[23], while in humans BRG1 was reported to facilitate XPC recruitment to damaged DNA[25]. However, in another study, a different role for human BRG1 in NER was

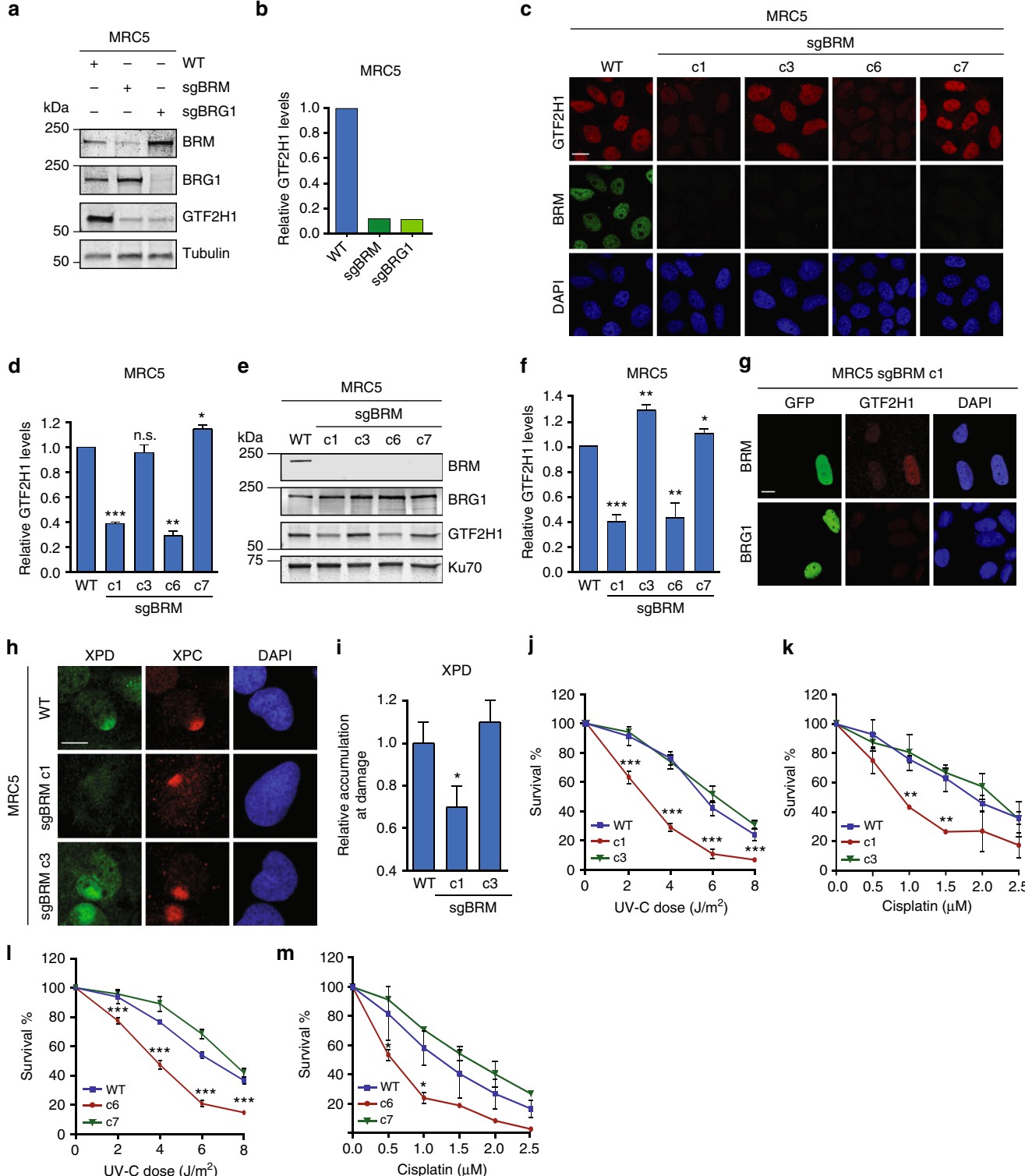

proposed, in facilitating XPG and PCNA—but not DDB2 and XPC—recruitment to sites of damaged DNA[24]. Our data, indeed, shows that both BRG1 and BRM are essential for efficient recruitment of late NER factors (TFIIH and downstream NER proteins) rather than for binding of the early DNA damage sensing factors (XPC, DDB2, and CSB) to DNA lesions. Impaired recruitment of the late NER factors in the absence of SWI/SNF is, however, not caused by reduced chromatin accessibility, but an indirect result of limiting amounts of functional TFIIH.

Furthermore, we did not observe BRM and BRG1 recruitment to UV-damaged DNA, further corroborating that SWI/SNF's main involvement in the UV-DDR is not in the control of chromatin accessibility at sites of UV damage.

SWI/SNF complexes are thought to be recruited to chromatin to remodel nucleosomes in enhancer and promoter regions to regulate transcription[7,46]. In line with this, we observed in two different cell types that BRM and BRG1 ChIP-seq signals are enriched at the *GTF2H1* promoter. SWI/SNF's influence on gene

**Fig. 6** DNA damage sensitivity of BRM-deficient cells correlates with GTF2H1 expression. **a** Immunoblot of BRM, BRG1, and GTF2H1 in MRC5 wild-type (WT) cells and cells transfected with sgRNA against BRM or BRG1. **b** Quantification of GTF2H1 levels in immunoblot shown in **a**, corrected by tubulin loading control, and set to 1.0 in MRC5 WT. **c** IF of total GTF2H1 and BRM levels in MRC5 WT and sgBRM knockout clones c1, c3, c6, and c7. Scale bar: 10 μm. **d** Quantification of GTF2H1 IF signal (shown in **c**). Total GTF2H1 levels were normalized to MRC5 WT, set to 1.0. Mean & S.E.M. of > 200 cells from two independent experiments. **e** Immunoblot of BRM, BRG1, and GTF2H1 levels in MRC5 WT and sgBRM clones c1, c3, c6, and c7. **f** Quantification of GTF2H1 levels shown in **e**, as described in **b**, using Ku70 as loading control. Mean & S.E.M. from four independent experiments. **g** GTF2H1 levels in a mixed population of MRC5 sgBRM knockout clone c1 cells non-transfected or transfected with BRM-GFP or BRG1-GFP. Scale bar: 5 μm. **h** XPD recruitment to LUD in MRC5 WT and sgBRM knockout clones c1 and c3, 30 min after damage. Scale bar: 5 μm. **i** Relative quantification of XPD recruitment to LUD (shown in **h**) in MRC5 WT and BRM knockout clones c1 and c3, normalized to WT, as described in the methods. Mean & S.E.M. of > 65 cells per sample. **j** UV-C colony survival of MRC5 WT cells and BRM knockout clones c1 and c3. **k** Cisplatin colony survival of MRC5 WT cells and BRM knockout clones c1 and c3. **l** UV-C colony survival of MRC5 WT cells and BRM knockout clones c6 and c7. **m** Cisplatin colony survival of MRC5 WT cells and BRM knockout clones c6 and c7. Colony number was normalized to untreated conditions. Mean & S.E.M. of four (UV-survival) and two (cisplatin-survival) independent experiments, each performed in triplicate, are presented. *$P < 0.05$, **$P < 0.01$, ***$P < 0.001$, relative to WT, n.s., non-significant. Full-size immunoblot scans are provided in Supplementary Fig. 7c, d

expression is, however, contextual, in that it represses some promoters while it stimulates others[5], which may also be evident from the differential effect of BRM knockdown on transcription of TFIIH genes that we observed. One major way through which SWI/SNF promotes transcription is by antagonizing the repressive activity of Polycomb complexes, as loss of SWI/SNF was shown to lead to repression of Polycomb target genes[47,48]. Nevertheless, we were unable to alleviate downregulation of GTF2H1 upon knockdown of BRM or BRG1 with specific inhibitors targeting EZH2, a functional enzymatic component of the Polycomb repressive Complex 2. This suggests that other mechanisms, possibly involving repressive chromatin structures or epigenetic marks, account for the diminished GTF2H1 expression.

Besides NER, SWI/SNF chromatin remodeling complexes are also involved in other DDR pathways[6,8,9], including regulation of DSB repair by non-homologous end-joining and/or homologous recombination[49,50]. It is, thus, likely that SWI/SNF mutations found in cancer contribute to increased genomic instability by disrupting multiple DDR pathways. As the majority of BRG1-deficient tumors are negative for mutations in other genes that can be targeted by existing therapies[40], it would be advantageous to exploit DDR deficiencies in SWI/SNF cancers therapeutically. Based on our analysis, one such DDR deficiency could be impaired NER due to downregulation of GTF2H1 expression, rendering SWI/SNF cancers more sensitive to DNA damaging chemotherapeutic drugs such as cisplatin (Fig. 7). However, we observed that in multiple established BRG1 and/or BRM-deficient cancer cell lines, GTF2H1 levels were not lowered, which is probably due to an, yet unknown, adaptation mechanism to compensate for the loss of BRM/BRG1 activity (Fig. 7). Previous studies showed partial mutual compensation between both ATPases[36,38,40]. Nevertheless, the fact that normal GTF2H1 levels were observed in cells lacking both BRG1 and BRM and that overexpression of BRG1 did not increase GTF2H1 levels in BRM-deficient cells suggests that BRM and BRG1 do not compensate for each other in regulating GTF2H1 expression. Our experiments with MRC5 BRM knockout cell lines confirm that cells can adapt to the loss of one of the SWI/SNF ATPases. Although knockout of BRM led to an initial overall reduction in GTF2H1 levels, after prolonged culturing and clonal selection we observed that many clones displayed normal GTF2H1 expression. Importantly, cells exhibited hypersensitivity to DNA damage induction by UV irradiation and cisplatin treatment only when GTF2H1 levels were low.

Recently, it was suggested that BRG1 expression could be used as a predictive biomarker for platinum-based chemotherapy response in NSCLC lines[51,52]. However, as we here demonstrate, sensitivity of SWI/SNF-deficient cells to DNA damaging agents such as cisplatin mainly depends on GTF2H1 expression levels. Therefore, reduced GTF2H1 expression may be a better predictive marker for platinum-drug sensitivity of SWI/SNF-deficient cancers (Fig. 7). Moreover, given the importance of TFIIH for transcription and repair, elucidating the mechanisms underlying SWI/SNF regulation of GTF2H1 expression and those that allow cells to adapt and restore GTF2H1 levels will be key to develop new strategies targeting SWI/SNF cancers. Such knowledge could potentially reveal how to revert the adaptation response to lower GTF2H1 levels, rendering SWI/SNF-deficient cells more susceptible to platinum drug chemotherapy.

## Methods

**Cell lines, culture conditions, and treatments**. U2OS (ATCC), SV40-immortalized human fibroblasts MRC5 (ATCC) and XP4PA[53] (XPC-deficient, with stable expression of XPC-GFP), XPCS2BA (XPB-deficient, with stable expression of XPB-GFP[28]) and CS1AN (CSB-deficient, with stable expression of GFP-CSB[54]) were cultured under standard conditions in a 1:1 mixture of DMEM (Lonza) and Ham's F10 (Lonza) supplemented with 10% fetal calf serum (FCS). C5RO primary fibroblasts (established in our laboratory) were cultured in Ham's F10 supplemented with 12% FCS; H1299 NSCLC (provided by Dr. Bert van der Horst), A549 NSCLC (provided by Dr. Suzan Pas), Hs 578T[55] breast cancer, A2780[38] ovarian cancer (provided by Corine Beaufort and Dr. John Martens), Hs 700T[36] and Panc-1[56] pancreatic cancer (provided by Dr. Bernadette van den Hoogen), SW13[36] adrenal cortex carcinoma and C33A[36] cervical carcinoma (provided by Dr. Jan van der Knaap) cells were cultured in a 1:1 mixture of DMEM (Lonza) and RPMI (Sigma) medium supplemented with 10% FCS. Stable XPC-GFP expressing XP4PA cells were generated using lentiviral transduction and selected with 0.3 μg/mL puromycin and FACS. Stable GFP-GTF2H1 expressing cells (A549, H1299) were generated using lentiviral transduction and selection with 5–10 μg/mL blasticidin. Stable BRM-GFP and BRG1-GFP expressing U2OS cells were generated using transfection and selection with 10 μg/mL Blasticidin. All cells were cultured in medium containing 1% penicillin-streptomycin at 37 °C and 5% $CO_2$. siRNA transfections were performed using RNAiMax (Invitrogen) 2 days before each experiment, according to the manufacturer's instructions. Plasmids transfections were performed using FuGENE 6 (Promega), according to the manufacturer's instructions. All cell lines were regularly tested for mycoplasma contamination.

**Plasmids, sgRNA, and siRNA**. Full-length human cDNAs of GTF2H1, BRG1 and BRM (a kind gift from Dr. Kyle Miller[57]), were fused to GFP and inserted into pLenti-PGK-Blast-DEST[58] to generate plasmids GFP-GTF2H1, BRG1-GFP and BRM-GFP. Full-length human XPC cDNA was fused to GFP and inserted into pLenti-CMV-Puro-DEST[58] to generate plasmid XPC-GFP. For the generation of knockout cell lines, sgRNA sequences targeting *BRM* (GTCTCCAGCCC-TATGTCTGG) and *BRG1* (CAGCTGGTTCTTGGTTAAATG) coding regions were cloned into pLenti-CRISPR-V1[59]. Cloning and plasmid details are available upon request. siRNA oligomers were purchased from GE Healthcare: CTRL (D-001210-05), BRM#1 (J-017253-06), BRM#2 (J-017253-07), BRG1 (L-010431-00), BRG1#2 (J-010431-06), BRG1#3 (J-010431-07), GTF2H1 (L-010924-00) and XPA (MJAWM-000011).

**UV-C irradiation**. UV-C irradiation was inflicted using a germicidal lamp (254 nm; TUV lamp, Phillips) with the indicated doses after washing cells with PBS. Local damage was generated using 60 J/m² of UV irradiation through an 8 μm polycarbonate filter (Millipore), as described in van Cuijk et al[60].

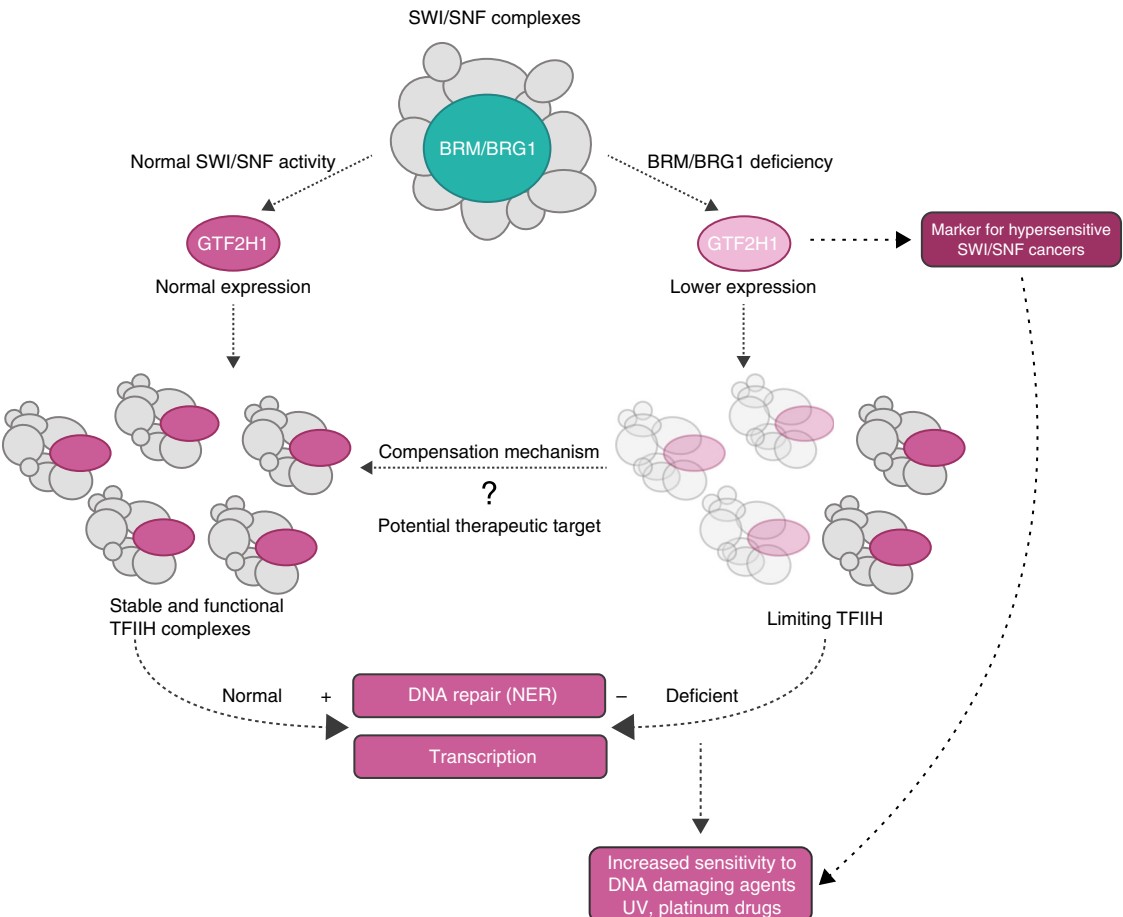

**Fig. 7** Low GTF2H1 expression as a predictive marker for DNA damage hypersensitive SWI/SNF cancers. BRM- and BRG1-containing SWI/SNF complexes promote the expression of the *GTF2H1* gene, a subunit of the TFIIH complex. In BRM- and BRG1-wild-type cells, normal expression of GTF2H1 allows the assembly of stable and functional TFIIH complexes, proficient in transcription and NER. When BRM or BRG1 are deficient, expression of GTF2H1 is lower, limiting the assembly of functional TFIIH complexes. As a consequence, transcription levels and NER capacity are lower, and cells become more sensitive to DNA damaging agents like UV and chemotherapeutic platinum drugs. Therefore, low GTF2H1 levels can likely be used as a marker to identify SWI/SNF cancers with increased sensitivity to chemotherapeutic drugs. However, cells with permanent loss of either BRM or BRG1 subunit can also adapt and restore the expression of GTF2H1, thus presenting normal transcription and NER activity. The mechanism underlying this adaption response is currently unknown, but if elucidated could be therapeutically exploited to specifically target SWI/SNF cancers with restored GTF2H1 expression, leaving surrounding non-tumor tissues intact

**Unscheduled DNA synthesis and RRS**. Fluorescent UDS and RRS were performed as described before[61]. In short, for UDS C5RO primary fibroblasts were grown on coverslips and treated with siRNAs 48 h before UV-C irradiation (16 J/m²). After irradiation, cells were incubated for 1 h in medium containing 5-ethynyl-2'-deoxyuridine (EdU, Invitrogen). For RRS, U2OS cells were seeded on coverslips and 48 h after siRNA transfection irradiated with 6 J/m² UV-C and allowed to recover for 2 or 20 h. Irradiated and non-irradiated cells were incubated for 2 h in medium containing 5-ethynyl-uridine (EU, Jena Biosciences). Cells were fixed in 4% paraformaldehyde and permeabilized with 0.1% Triton X-100 in PBS. EdU or EU incorporation was visualized by incubating cells for 1 h at room temperature with Click-it reaction cocktail containing Atto 594 Azide (60 μM, Atto Tec.), Tris-HCl (50 mM, pH 7.6), CuSO₄*5H₂O (4 mM, Sigma) and ascorbic acid (10 mM, Sigma). After washes in 0.1% Triton-X100 in PBS, DNA was stained with DAPI (Sigma), and slides were mounted using Aqua-Poly/Mount (Polysciences, Inc.). Images were acquired using an LSM700 microscope equipped with a 40x Plan-apochrat 1.3 NA oil immersion lens (Carl Zeiss Micro Imaging Inc.). UDS and RRS levels were quantified by measuring the total nuclear fluorescence intensities (in at least 100 cells per experiment) with FIJI image analysis software. Intensity levels were averaged and normalized to the fluorescence levels in control conditions, which were set at 100%.

**Immunofluorescence**. Cells were grown on coverslips, fixed in 4% paraformaldehyde and permeabilized in PBS containing 0.5% Triton X-100. DNA was denatured for 5 min with 70 mM NaOH to allow CPD binding by the antibody. Next, cells were incubated for 1 h with blocking solution 3% BSA in PBS-T (0.1% Tween 20) and subsequently incubated with antibodies diluted in 1% BSA with

PBS-T (0.1% Tween 20) for 1–2 h at room temperature or overnight at 4 °C. To visualize primary antibodies, cells were incubated for 1 h at room temperature with secondary antibodies conjugated to Alexa fluorochromes 488, 555, or 633 (Invitrogen). DNA was stained with DAPI (Sigma), and slides were mounted using Aqua-Poly/Mount (Polysciences, Inc.). Antibodies used are summarized in Supplementary Tables 1 and 2. Images were acquired using an LSM700 microscope equipped with a 40x Plan-apochromat 1.3 NA oil immersion lens (Carl Zeiss Micro Imaging Inc.). Using FIJI image analysis software, we determined protein accumulation at lesion sites by dividing the overall fluorescence signal intensity at LUDs by the protein overall nuclear intensity. In Fig. 1f and Fig. 6g zero accumulation (nuclear background) was set at 0 and maximum accumulation (above nuclear background) in control conditions at 1.0.

**Fluorescence recovery after photobleaching (FRAP)**. FRAP experiments were performed as previously described[60,62], using a Leica TCS SP5 microscope (with LAS AF software, Leica) equipped with a 40×/1.25 NA HCX PL APO CS oil immersion lens (Leica Microsystems), at 37 °C and 5% CO₂. Briefly, a strip spanning the nucleus width (512 × 16 pixels) at 1400 Hz of a 488 nm laser, with a zoom of 12x was used to measure the fluorescence signal every 100 ms until a steady-state was reached (pre-bleach). Fluorescence signals were then photo-bleached using 100% power of the 488 nm laser and recovery of fluorescence in the strip was monitored every 22 ms until a steady-state was reached. Fluorescence signals were normalized to the average pre-bleach fluorescence after background signal subtraction. Three independent experiments were performed, with the acquisition of ten cells for each condition in each experiment. The immobile fraction ($F_{imm}$), shown in Fig. 2c, was determined using the fluorescence intensity

recorded immediately after bleaching ($I_0$) and the average fluorescence signal after reaching steady-state from the unchallenged cells ($I_{final,unc}$) and UV-irradiated cells ($I_{final,UV}$):[60]

$$F_{imm} = 1 - \frac{I_{final,UV} - I_{0,UV}}{I_{final,unc} - I_{0,UV}}.$$

**Real-time protein recruitment to UV-C laser-induced damage**. To induce local UV-C DNA damage in living cells, a 2 mW pulsed (7.8 kHz) diode pumped solid state laser emitting at 266 nm (Rapp Opto Electronic, Hamburg GmbH) coupled to a Leica TCS SP5 confocal microscope was used, as described previously[61]. Cells seeded on quartz coverslips were imaged and irradiated via a Ultrafluar quartz 100×/1.35 NA glycerol immersion lens (Carl Zeiss Micro Imaging Inc.) at 37 °C and 5% $CO_2$. Resulting accumulation curves were corrected for background values and normalized to the relative fluorescence signal before local irradiation.

**Chromatin fractionation**. U2OS cells were grown to confluency on 10 cm dishes, UV-C irradiated with the indicated dose and lysed in lysis buffer (30 mM HEPES pH 7.6, 1 mM $MgCl_2$, 130 mM NaCl, 0.5% Triton X-100, 0.5 mM DTT and EDTA-free protease inhibitor cocktail (Roche)), at 4 °C for 30 min. Non-chromatin bound proteins were recovered by centrifugation (10 min, 4 °C, 16,100 g). Chromatin-containing pellet was resuspended in lysis buffer supplemented with 250 U/μL of Benzonase (Merck Millipore) and incubated for 1 h at 4 °C. Equal amounts of sample were used for SDS-PAGE gels and immunoblotting analysis.

**Cycloheximide (CHX) protein stability assay**. Protein synthesis was inhibited by adding 100 μM CHX (Enzo) to cells in culture. Concomitantly, for the experiment shown in Supplementary Fig. 3a, protein degradation was inhibited by adding 10 μM MG132 (Sigma) before the addition of CHX. Cells were lysed at the indicated time points after CHX addition, for 30 min at 4 °C in RIPA buffer (25 mM Tris-HCl pH 7.5, 150 mM NaCl, 6 mM EDTA, 0.1% SDS, 1% Triton X-100, 1% NP-40, supplemented with EDTA-free protease inhibitor cocktail (Roche)). Whole cell extracts were recovered by centrifugation (20 min at 4 °C and 1400 g) and quantified using the BCA Protein Assay Kit (Pierce, ThermoFisher Scientific). Equal amounts of protein from total cell lysates were used for immunoblot analysis.

**Immunoblotting**. Protein samples (whole cell extracts or cell fractionations) were 2 x diluted in sample buffer (125 mM Tris-HCl pH 6.8, 20% Glycerol, 10% 2-β-Mercaptoethanol, 4% SDS, 0.01% Bromophenol Blue) and boiled for 5 min at 98 ° C. Equal amounts of protein from whole cell lysates were separated in SDS-PAGE gels and transferred onto PVDF membranes (0.45 μm, Merck Millipore). After 1 h of blocking in 5% BSA in PBS-T (0.05% Tween 20), membranes were incubated with primary antibodies in PBS-T for 1–2 h at room temperature, or at 4 °C overnight. Secondary antibodies were incubated for 1 h at room temperature. Membranes were washed 3 × 10 mins in PBS-T after antibody incubation. Probed membranes were visualized with the Odyssey CLx Infrared Imaging System (LI-COR Biosciences). Antibodies are listed in Supplementary Table 1 and 2. Immunoblots were quantified using ImageStudio Lite (ver. 5.2, LI-COR Biosciences). Full-size immunoblot scans are provided in Supplementary Fig. 6,7.

**Colony forming assays**. For colony survival assays after DNA damage, cells were seeded in triplicate in six-well plates (400 cells/well) and treated with increasing doses or concentrations of UV-C or cisplatin, respectively, 1 day after seeding. After 5–7 days, colonies were fixed and stained. For the colony forming assay shown in Fig. 5d,e and Supplementary Fig. 4e, cells were seeded in triplicate in six-well plates (750–1000 cells/well) 48 h after siRNA transfection. After 12 days, cells were fixed and stained. Fixing and staining solution: 0.1% w/v Coomassie Blue (Bio-Rad) was dispersed in a 50% Methanol, 10% Acetic Acid solution. Colonies were counted with the integrated colony counter GelCount (Oxford Optronix).

**Real-time reverse transcriptase PCR (RT-qPCR)**. Total RNA was isolated from siRNA-transfected U2OS cells using the RNeasy mini kit (Qiagen). cDNA was synthesized using iScript cDNA Synthesis Kit (Bio-Rad), accordingly to manufacturer's instructions. TFIIH genes and GAPDH expression levels were analyzed using RT-qPCR with the PowerUP SYBR Green Master Mix (ThermoFischer Scientific) in a Bio-Rad CFX96 device. Primers used are listed in Supplementary Table 3. The relative gene expression of TFIIH genes was calculated according to the comparative quantification cycle (Cq) method and normalized to *GAPDH* expression. The expression level of each TFIIH gene in BRM knockdown cells was normalized to expression in control siRNA treated cells. Expression levels were measured in triplicate in two independent experiments.

**Re-analysis of public Chip-seq data**. To dissect BRG1/BRM enrichment in GTF2H1, we re-analyzed published BRG1/BRM ChIP-seq datasets from liver cancer HepG2 cells upon transfection with non-targeting shRNA (Fig. 3c; GEO accession GSE102559[32]) and BRG1/BRM ChIP-seq datasets from RWPE1-SCHLAP1 cells (Supplementary Fig. 3c; GEO accession GSE114392[33]). ChIP-seq raw data was obtained from the Sequence Read Archive repository (SRA, NCBI; SRP115303 and SRP145601) and uploaded to the Galaxy platform[63]. Reads were

aligned to the human genome (hg19 build) with BWA (Galaxy Version 0.7.17.4), poor quality alignments and duplicates were subsequently filtered with SAMtools (Galaxy Version 1.1.2) –q 20. To visualize ChIP-seq signal density, replicate datasets were merged with SAMtools and further processed using bamcoverage tool (Galaxy Version 2.5.0.0), DeepTools suit[64] with binsize 30, reads extended to 150 bp and normalized to reads per kilobase per million (RPKM); resulting bigwig files were visualized using IGV genome browser[65]. Peaks were determined with MACS2 peak caller (Galaxy Version 2.1.1.20160309.0[66]) using the *predictd* function to estimate fragment size for all datasets and the following analysis parameters –qval = 0.01 –nomodel –extsize = d –broad -broadcutoff 0.05 –keepdup-all. Resulting peaks were filtered against the ENCODE blacklist regions and finally visualized in IGV browser. Promoter region annotation for *GTF2H1* gene was obtained from the Ensembl database (GRCh37 assembly, Chr11: 18,340,602–18,346,999).

**Immunoprecipitation**. The procedure for in vivo crosslink and immunoprecipitation was described previously[12] and applied with minor alterations. Briefly, after UV-C irradiation (20 J/m²), cells were cultured for 30 min before crosslinking in 1% paraformaldehyde in PBS for 5 min at room temperature. Crosslinking reaction was stopped with 0.125 M of glycine and cells were collected in ice cold PBS supplemented with 1 mM PMSF and 10% glycerol. All subsequent steps were performed at 4 °C. Following centrifugation, cell pellet was resuspended in lysis buffer (50 mM HEPES pH 7.8, 0.15 M NaCl, 0.5% NP-40, 0.25% Triton X-100, and 10% glycerol). After 30 min incubation, the suspension was spun down, and supernatant (soluble fraction) was removed. The pellet was washed with Wash buffer (0.01 M Tris-HCl pH 8.0, 0.2 M NaCl), spun down and resuspended in 1 × RIPA buffer (0.01 M Tris-HCl pH 7.5, 0.15 M NaCl, 1% Triton X-100, 1% NP-40, 0.1% SDS). Chromatin was sheared using a Bioruptor Sonicator (Diagenode) using cycles of 30 s ON, 30 s OFF during 10 min, after which samples were centrifuged. The supernatant containing crosslinked chromatin was used for immunoprecipitation. All buffers were supplemented with 0.1 mM EDTA, 0.5 mM EGTA, 1 mM PMSF and a mixture of proteinase and phosphatase inhibitors. For immunoprecipitation, extracts were incubated with GFP-trap beads (Chromotek), overnight at 4 °C. Subsequently, beads were washed five times in RIPA buffer and elution of the precipitated proteins was performed by extended boiling in 2x Laemmli sample buffer for immunoblotting analysis.

**Cell cycle profiling**. For cell cycle analysis, cells were fixed in 70% ethanol, followed by DNA staining with 50 μg/mL propidium iodide (Invitrogen) in the presence of RNase A (0.1 mg/mL). Cell sorting was performed on a BD LSRFortessa™ flow cytometer (BD Bioscience) using FACSDiva software. Obtained data was quantified with Flowing software 2.5.1 (by Perttu Terho in collaboration with Turku Bioimaging).

**Statistical analysis**. Mean values and S.E.M. error bars are shown for each experiment. Unpaired, two-tailed *t* tests were used to determine statistical significance between groups. In all experiments, between-group variances were similar and data were symmetrically distributed. For analysis of graphs in Fig. 2a and Supplementary Fig. 1g, a ROC curve analysis was performed with significance levels set to 0.05. All analysis were performed using Graph Pad Prism version 7.03 for Windows (GraphPad Software, La Jolla California USA). P values expressed as *$P < 0.05$; **$P < 0.01$, ***$P < 0.001$ were considered to be significant. n.s, non-significant.

## Data availability

The raw ChIP-seq data sets analyzed during the current study were obtained via the Sequence Read Archive repository (SRA, NCBI), [https://www.ncbi.nlm.nih.gov/sra], with the data set identifiers SRP115303 and SRP145601. Other relevant data generated during the current study are available from the corresponding author on reasonable request. Individual data points are provided in Supplementary Data 1.

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

## Acknowledgements

The authors would like to thank Dr. Serena Bruens and Dr. Akos Gyenis for advice, reagents and technical assistance with RT-qPCR experiments; Dr. David Cano and Mireille van de Veer for technical assistance and the Erasmus MC Optical Imaging Center for microscopy support. The CCNH antibody clone 2D4 was kindly provided by Dr. Jean-Marc Egly, pcDNA6.2-BRM-emGFP plasmid by Dr. Kyle Miller. We thank Dr. Suzan Pas, Dr. Bernadette van den Hoogen, Corine Beaufort, Dr. John Martens, Dr. Bert van der Horst and Dr. Jan van der Knaap for BRG1 and BRM-deficient cells. This work was supported by a Marie Curie Initial Training Network funded by the European Commission 7th Framework Programme (grant 316390), a European Research Council Advanced Grant (grant 340988-ERC-ID), a Worldwide Cancer Research Award (grant 15-1274), Dutch Scientific Organization (ALW grants 854.11.002 and 864.13.004). This work is part of the Oncode Institute which is partly financed by the Dutch Cancer Society and was funded by a grant from the Dutch Cancer Society (KWF grant 10506).

## Author contributions

C.R.S., H.L., J.S. and A.H. performed all experiments. O.Z.A. initiated this study and contributed reagents. R.M.R., J.A.M. and J.H.J.H. analyzed data and advised. C.R.S., H.L. and W.V. designed experiments, analyzed data and wrote the manuscript. All authors reviewed the manuscript.

## Additional information

**Competing interests:** The authors declare no competing interests.

