## [Peer Review File · Nature Communications]

Reviewers' comments:

Reviewer #1 (Remarks to the Author):

In this manuscript, Ribeiro-Silva et al. provide evidence for a role of the SWI/SNF ATPase subunits BRM and BRG1 in the NER pathway, specifically in impairing transcription and expression of TFIIH subunit GTF2H1. The role of SWI/SNF in NER is not novel, and has been controversial, as reports in the literature have suggested it functions in recruiting XPC to sites of UV crosslinks, while others identify a role in recruiting downstream factors such as XPG to sites of UV damage. Here, the authors report that BRM and BRG1 indirectly act on the NER pathway, downstream of initial lesion detection steps by regulating transcription of GTF2H. The authors further extend on previously reported findings of synthetic lethality between BRM and BRG1 and that moreover GTFH1 levels are predictive of cisplatin and UV sensitivity. While a finding that BRM and BRG1 regulates transcription of GTFH1, which promotes NER and is important for cisplatin and UV sensitivity is potentially interesting, the overall conceptual advance is relatively incremental over previous publications linking BRM, BRG1, and GTF2H1 to NER and cancer treatment response and the authors' conclusions are not fully supported their data due to concerns of interpretation and incomplete experimental design.

Main comments:

1. The authors' conclusions of a transcription dependent function for BRG1 in promoting NER are at odds with reports by Zhang, Cell Cycle, 2009 and Zhao, JBC, 2009 that BRG1 localizes to DNA damage sites and interacts with XPC. If BRG1 is promoting NER through transcription, why would it localize to DNA damage sites and interact with components of the NER?
2. Most of the experiments in this study are completed with only 1 siRNA and since a major conclusion of the report is that BRG1/BRM promote transcription of GTFH1, it is possible that this is through an off-target effect. To provide confidence for these findings, the authors should show each of their phenotypes with at least 2 siRNAs and ideally rescue with WT and catalytically inactive mutants. Furthermore, if BRG1 and BRM promote NER through GTFH1, is there an epistatic relationship of BRM and GTFH1 knockdown or of BRG1 and GTFH1 knockdown?
3. Authors speculate that BRM1/BRG can compensate for each other mechanistically but do not present data to support this claim. Can overexpression of BRM1 compensate for BRG-depleted impairment of GTF2H1 transcription? Can the compensatory re-expression of GTF2H1 in certain clones in BRG KO cell lines be transiently rescued by knock down of BRM1?
4. Could some of phenotypes be attributed to indirect effects of BRG1 and BRM on cell cycle?
5. Fig. 3A – differences in half-life of XPB do not appear as strong- 24 > 6h as claimed by authors and seem comparable to that of TFIIEB. Better quality western blot would give more confidence to this conclusion.
6. Fig. 4A – the rescues are not convincing. It appears that there is XPD recruitment even when GTF2H1 is expressed. The authors should provide quantitation as well as a western demonstrating comparable expression of rescue in the cells used.
7. Fig. 5 – it is claimed that BRG1 knockdown only lowers GTFH1 in U2OS but not A549 and H1299 cells; however, due to the low baseline GTFH1 levels, this claim is insufficiently supported. The authors should validate knockdown in these cells in another manner perhaps by qRT-PCR.
8. Fig. S4B – CRISPR/Cas9 does not appear to be knocking out BHM expression. The authors claim that there is decreased levels in H1299 but this not supported by their data. The authors should validate genetic knockout in all alleles.

9. Fig. 5 – levels of GTFH1 should be shown by western blot analysis rather than IF which is less quantitative and more subjective to interpretation of expression levels.

10. Authors should expand on their key finding that BRM/BRG1 regulates transcription of GTF2H1. At minimum, CHIP-Seq experiments should be performed to validate that SWI/SNF is acting specifically at this locus in response to UV. (Fig. 3)

11. Is there a correlation between loss of GTF2H1 and BRM/BRG1 deficiency in cancer patients? And if so, does this deficiency translate to cisplatin sensitivity? Mining data from the cancer therapeutics response portal would be useful to address this.

12. Fig 5c-e: dose response curves should be shown, not just data from one dose.

Minor Comments:

1. Fig 2B – needs nucleoplasmic marker

2. There is reference to a Fig. 4C in the figure legend but no corresponding pic or reference in the Results section.

3. There is incorrect referencing of figures – i.e. “Depletion of BRG1 led to lower overall transcription (Fig. 3E)”

4. Most figures contain error bars, but importantly, lack statistical analysis: see Fig. 1a,c; 2a,c,e; 3b,c,e; 5d,e; 6b,h,i. Moreover, a number of studies were only completed with 2 replicas.

5. Fig S5c : Western data for sgBRM clones c6 and c7 should be shown.

Reviewer #2 (Remarks to the Author):

This paper by Ribeiro-Silva and colleagues elaborates on the impact of SWI/SNF ATPases on DNA damage sensitivity. In their study the authors demonstrate that BRM and BRG1 facilitate the expression of GTF2H1. This functional relationship is examined in selected cancer cell lines where the authors observe that, despite the loss of the ATPases, GTF2H1 levels are restored. The authors speculate that this might open an avenue for therapeutic intervention.

The functional relationship between the SWI/SNF and GTF2H1 is interesting but also quite trivial. SWI/SNF complexes have a broad spectrum of target genes and it is not surprising that DNA repair factors are amongst them. What about other SWI/SNF target genes that might cause the observed phenotypes in cancer cells? Just to give an example, I believe a genome-wide view would have made a stronger point than selecting NER target genes for qPCR analysis. There is no clear indication of a molecular mechanism, which in my opinion would be mandatory to publish this paper in Nature Communications. Does BRM/BRG1 localize to the GTF2H1 locus? Does BRM/BRG1 have a general impact on the chromatin conformation (ATAC-seq)? Eventually, any therapeutic intervention demands not only detailed insight into mechanism but also needs to address potential off-target effects. How do cells respond to the depletion of the ATPases or GTF2H1 ? What about their chromatin conformation? How is the general transcription efficacy affected ? Similarly, I find the examination of cancer-related phenotypes rather preliminary and not convincing. A pure analysis of protein expression levels and one colony formation experiment cannot make a strong claim.

I realize that a lot of work has gone into this research project but I feel that the paper in its current form is rather descriptive and reveals correlations rather than demonstrating clear-cut and

convincing data. For a high impact journal like Nature Communications I expect a more detailed analysis and, equally important, a novel and intriguing mechanism.

Minor comments

1. In Figure 1 and Figure 2B the knockdown levels of the protein have neither been analyzed by Western blotting nor microscopically.
2. In Figure 2D the XPD and XPB levels do not increase at chromatin but the levels diminish in the nucleoplasm. Figure 3D shows that knockdown of BRM1 has no impact on the global XPD and XPB levels. How can diminished levels of the proteins in the nucleoplasm be explained?
3. The quality of many western blots is below standard.
4. The qPCR analysis on TFIIH subunits is not sufficient. Does BRM act directly on these genes (ChIP experiments) or are these indirect effects?
5. Figure 4 does not have any quantification nor statistical analysis.
6. It is stated that BRG1 and BRM may compensate for each others functions. This should be tested.
7. The phenotype when assessing the CRISPR mediated knockdown of BRM and BRG1 is not convincing. I would suggest to conduct a more detailed analysis.

Reviewer #3 (Remarks to the Author):

SWI/SNF chromatin remodeling complexes are highly mutated in cancers. Great efforts are underway to understand their function and their possible involvement in the carcinogenic process. In their manuscript "DNA damage sensitivity of SWI/SNF-deficient cells depends on TFIIH subunit p62/GTF2H1" Ribeiro-Silva et al tackle this challenge by addressing the role of the two ATPase subunits of SWI/SNF complexes in nucleotide excision repair (NER).

Their major findings:

- 1) The authors performed knock down for each of the ATPases of SWI/SNF, BRG1 and BRM. They then conducted unscheduled DNA synthesis and recovery of RNA synthesis experiments in these knock-down cells and showed that NER of UV damage was compromised.
- 2) Since there have been previous works on the role of BRG1 in NER, the authors focused their work to the BRM ATPase and showed that after BRM siRNA post UV recruitment to chromatin of XPC and DDB2 was unaffected, however the downstream recruitment of TFIIH subunits XPB, XPD and GTF2H1, and recruitment of XPA and XPF decreased.
- 3) Chromatin binding of TFIIH was reduced even in the absence of damage. There was no evidence for direct interaction between BRM and TFIIH and BRM is not recruited to chromatin after damage.
- 4) Assaying protein and RNA expression after BRM knock down showed that expression of the TFIIH subunit GTF2H1 was significantly decreased. XPB levels were mildly decreased, but the protein was clearly destabilized. Together, this pointed to GTF2H1 as the primary reason for TFIIH destabilization and reduced chromatin binding in BRM knock down. Indeed, GTF2H1 knock down resulted in low RNA synthesis.
- 5) Cancer cells with BRG1 defects have normal GTF2H1 levels - which can be reduced by BRM knock down. This led the authors to suggest a compensation mechanism -which could explain the synthetic lethality of BRM knock-out. However, since ectopic expression of GTF2H1 could not prevent this synthetic lethality, it is clear that GTF2H1 is not the sole driver of the deleterious effects of BRG1/BRM deficiencies.
- 6) Growing clones of BRM/BRG1 knock out cells showed that indeed, with time, certain clones express GTF2H1 at normal levels, and the different levels of GTF2H1 indeed confer different UV and Cisplatin sensitivities.

In general, I find this to be an important paper that initially set out to investigate the role of BRM in DNA repair - and found that the role of BRM in repair is primarily a byproduct of its role in transcription. They uncover a mechanism by which mutations in BRM and BRG1 alter transcription levels in cells and as a result, the ability to repair DNA. Understanding the function of SWI/SNF proteins, that are highly mutated in cancer is extremely important both to understanding their contribution to cancer risk and to identify possible cancer-specific targets for therapy.

Major comments:

- 1) Previous reports indicated that BRG1 was involved in repair of CPD and not (6-4)PP - indicating it may be specifically involved in TC-NER and not GG-NER (Zhao Q. et al, JBC 2009, Gong F. et al, Cell Cycle 2008, DOI: 10.4161/cc.7.8.5647). The authors measure repair using the unscheduled DNA synthesis assay that measures total NER across the genome, and see a reduction after knock down of BRM and BRG1. They draw the conclusion that GG-NER is therefore affected. However, for the major UV damage - CPD - repair in early time points is primarily transcription coupled. Therefore, theoretically, even if BRM/BRG1 knock down solely affected TCR would you not see this effect? Could you also knock down CSB and show that in the absence of CSB BRG/BRM knock down also affects repair - showing it really is affecting GG-NER?
- 2) The results indicate that BRM primarily affects repair through its stabilizing of TFIIH - and GTF2H1 expression. Since it does not interact directly with TFIIH - one possible mechanism would be by enhancing its transcription (binding the GTF2H1 promoter). ChIP of these subunits may be tricky, but there are of genome-wide ChIP-seq data sets available that could be useful such as doi: 10.1186/s13072-017-0167-8

Minor comments:

- 1) Figure 1: For each siRNA condition, the untreated cells are considered as 100%. Since BRG1 and BRM result in a general reduction of RNA synthesis, would it not be a more accurate representation of the to normalize all samples to untreated siControl?
- 2) Figure 3: First, a suggestion to change the order of the figures - to first show RNA and protein expression is lower, and then XPB stability is lower. Second, I think in Fig3b: I would name the plot XPB stability and not turnover, because turnover was not specifically measured, rather the protein levels.
- 3) I would like to see quantifications of figure 4 since the effects are partial and not complete.
- 4) Figure 6c shows the different levels of GTF2H1 in the two clones assayed. However, it is important to show that indeed in the same cells BRM is indeed knocked out. This data is currently in supplemental figure 5.
- 5) Figure 7: Could the general lower RNA expression in TFIIH deficient cells not further reduce BRM/BRG1 expressio - leading to an additional negative feedback?

We would like to thank all three reviewers for their generally constructive criticism and suggestions to improve our manuscript. Please find below our point-to-point response to the reviewer's comments.

Reviewer #1 (Remarks to the Author):

In this manuscript, Ribeiro-Silva et al. provide evidence for a role of the SWI/SNF ATPase subunits BRM and BRG1 in the NER pathway, specifically in impairing transcription and expression of TFIIH subunit GTF2H1. The role of SWI/SNF in NER is not novel, and has been controversial, as reports in the literature have suggested it functions in recruiting XPC to sites of UV crosslinks, while others identify a role in recruiting downstream factors such as XPG to sites of UV damage. Here, the authors report that BRM and BRG1 indirectly act on the NER pathway, downstream of initial lesion detection steps by regulating transcription of GTF2H. The authors further extend on previously reported findings of synthetic lethality between BRM and BRG1 and that moreover GTFH1 levels are predictive of cisplatin and UV sensitivity. While a finding that BRM and BRG1 regulates transcription of GTFH1, which promotes NER and is important for cisplatin and UV sensitivity is potentially interesting, the overall conceptual advance is relatively incremental over previous publications linking BRM, BRG1, and GTF2H1 to NER and cancer treatment response and the authors' conclusions are not fully supported their data due to concerns of interpretation and incomplete experimental design.

We would like to thank this reviewer for his/her concerns, which allowed us to strengthen our main findings by repeating and adding several experiments and controls. We are confident that with additional new data and textual revisions, we will take away the concerns over interpretation or incomplete experimental design. Also, we would like to emphasize that although BRG1 has been implicated in NER before, its exact involvement has been a matter of debate and conflicting models for its activity have been put forward. Moreover, BRM was never functionally or mechanistically linked to NER and there are no 'previous publications linking BRM, BRG1, and GTF2H1 to NER and cancer treatment response'. Thus, we do not understand why our findings would be 'relatively incremental' over these non-existing previous publications.

Main comments:

1. The authors' conclusions of a transcription dependent function for BRG1 in promoting NER are at odds with reports by Zhang, Cell Cycle, 2009 and Zhao, JBC, 2009 that BRG1 localizes to DNA damage sites and interacts with XPC. If BRG1 is promoting NER through transcription, why would it localize to DNA damage sites and interact with components of the NER?

We are aware of these previous publications about BRG1, which we also cited and discussed with respect to our data in the discussion section. The Zhao paper is in line with our results, showing impaired recruitment of XPG, PCNA and XPB (visible in Fig. 2c of this paper) but not of DDB2 and XPC after siBRG1. The Zhang paper contrasts with the Zhao paper and reports lower XPC recruitment after BRG1 depletion. Importantly, in our manuscript, we have not studied the recruitment of XPC after BRG1 depletion. These previous results are therefore not necessarily at odds with ours. We did study the recruitment of XPC after BRM depletion (using multiple complementary techniques such as immunofluorescence, live cell imaging and cell fractionation) and did not find evidence of XPC recruitment being affected by BRM.

In addition, we did not find any evidence of BRG1 or BRM localization to UV-damaged sites. In our original manuscript, we already addressed this using the same technique as in the Zhang paper (shown in Supplementary Fig. 2b), showing no recruitment of BRG1 (and BRM)

to local UV damage inflicted through a microporous membrane (at time points at which XPC is also recruited). Cellular fractionation also showed no enrichment of BRM in chromatin after UV (Fig. 2d). To confirm this, in the revised manuscript, we have additionally tested recruitment of fluorescently tagged BRG1 (and BRM) to DNA damage sites induced by 266 nm UV-C laser. This is a more powerful and sensitive technique than microporous membrane-mediated local damage infliction and was developed in our lab (Dinant et al, J Cell Science, 2007). This technique allows real-time visualization, even of transient recruitment and of TC-NER factors (Dinant et al, Mol Cell, 2013; Aydin et, Nucleic Acids Res, 2014; Schwertman et, Nature Genetics, 2012), which is not possible using a microporous membrane. As shown in the new Supplementary Fig. 2c, also with this technique, we did not observe any re-localization of BRG1 (and BRM) to UV-induced DNA damage. Thus, we are confident that these ATPases are not recruited to local UV-damage. Therefore, we cannot speculate on BRG1 localization to DNA damage sites.

2. Most of the experiments in this study are completed with only 1 siRNA and since a major conclusion of the report is that BRG1/BRM promote transcription of GTFH1, it is possible that this is through an off-target effect. To provide confidence for these findings, the authors should show each of their phenotypes with at least 2 siRNAs and ideally rescue with WT and catalytically inactive mutants. Furthermore, if BRG1 and BRM promote NER through GTFH1, is there an epistatic relationship of BRM and GTFH1 knockdown or of BRG1 and GTFH1 knockdown?

We understand the common concern of off-target effects by siRNA experiments of the reviewer. However, we have provided multiple lines of evidence shown in our original manuscript to rule out off-target effects (unless 5 different si/sgRNA sequences all have the same off-target effect):

- Both independent siRNAs against BRM and BRG1 lead to the same phenotype (lower GTF2H1 levels causing dysfunctional NER). It is highly unlikely that both siRNAs targeting different genes have the same off-target effect.
- XPB recruitment to UV-C laser was shown to be diminished using two independent siRNAs against BRM (Fig. 2a,b and Supplementary Fig. 1g). As stated in the text, this experiment was included to exclude siRNA off-target effects for siBRM.
- The phenotype observed after siBRM and siBRG (i.e. lowered GTF2H1 levels) was also shown using independent sgRNAs targeting BRM and BRG1 (Fig. 6a-f, Supplementary Fig. 5a). Moreover, reduced GTF2H1 by sgRNA-mediated knockout of BRM was rescued by ectopic BRM expression (Fig. 6g), demonstrating that the effect is mediated by BRM itself.
- GTF2H1 levels were not reduced with siBRG1 in BRG1-deficient cancer cells (Fig. 5a,b), demonstrating that the effect of siBRG1 is mediated by BRG1 and not an off-target protein/RNA.
- Depletion of BRM or BRG1 does not reduce levels of ectopically expressed GFP-tagged GTF2H1, while this does reduce in the same cells endogenous GTF2H1 (Fig. 5c, Supplementary Fig. 5c,d). This indicates that siRNAs against BRM and BRG1 do not act on GTF2H1 mRNA directly.

Despite this wealth of evidence, we still performed the requested experiments and included new figures showing clear reduction in GTF2H1 levels following additional, independent siRNAs against BRM and against BRG1 (Supplementary Fig. 3a,b,g,h). Thus, we now show in the revised manuscript that with two independent siRNAs and with an independent sgRNA, both for BRM and for BRG1, that the levels of GTF2H1 are reduced.

To prove that BRG1 and BRM promote NER through GTF2H1, we showed that the defect in TFIIH (XPD) recruitment to local sites of UV damage after siBRG1/siBRM could be rescued by overexpressing GTF2H1, for which we now also include the quantification (Fig. 4a-c). To confirm that there is an epistatic relationship between BRG1/BRM and GTF2H1, we added

an experiment to the revised manuscript in which we tested recruitment of XPD following double siRNA treatment against GTF2H1 and either BRG1 or BRM (Supplementary Fig. 3k). This experiment shows no additional reduction in XPD recruitment to UV damage by siBRG1 or siBRM after GTF2H1 depletion, indicative of an epistatic relationship.

3. Authors speculate that BRM1/BRG can compensate for each other mechanistically but do not present data to support this claim. Can overexpression of BRM1 compensate for BRG-depleted impairment of GTF2H1 transcription? Can the compensatory re-expression of GTF2H1 in certain clones in BRG KO cell lines be transiently rescued by knock down of BRM1?

Our discussion that BRG1 and BRM might compensate for each other's function was based on literature in which such compensation was reported with regard to other functions of BRG1/BRM (for instance, see Strobeck et al, J. Biol. Chem, 2002; Hoffman et al, PNAS, 2014; Oike et al, Cancer Research, 2013). Thus, this seemed like a plausible mechanism. We have now tested whether indeed ectopic expression of BRG1 can compensate for the loss of BRM, either by siRNA or in BRM sgRNA knockout clones. Both experiments, now shown in Fig. 6g and Supplementary Fig. 5d, indicate that with regard to GTF2H1 regulation, BRG1 does not compensate for BRM. We have described this novel evidence clearly in the results and rephrased our results and discussion section of the revised manuscript.

4. Could some of phenotypes be attributed to indirect effects of BRG1 and BRM on cell cycle?

We thank the reviewer for this thoughtful question. However, NER is not differentially active throughout the cell cycle. Thus, it is not expected that the observed phenotypes are due to indirect cell cycle effects. However, we now include new experiments in the revised manuscript, showing that depletion of BRM or BRG1 does not impact the cell cycle (Supplementary Fig. 3j).

5. Fig. 3A – differences in half-life of XPB do not appear as strong- 24 > 6h as claimed by authors and seem comparable to that of TFIIIEB. Better quality western blot would give more confidence to this conclusion.

Although the presented quantifications of the XPB half-life were obtained using unbiased ImageStudio (LI-COR Biosciences) image software analysis, we have repeated the experiments shown in Fig. 3a,b and Supplementary Fig. 3b of the original manuscript. We included new and better quality western blots in the revised manuscript (Fig. 3d) and have now quantified XPB and TFIIIEB levels from multiple independent replicate experiments, again clearly showing accelerated turnover of XPB but not of TFIIIEB after BRM or GTF2H1 depletion (Fig. 3d,e and Supplementary Fig. 3e).

6. Fig. 4A – the rescues are not convincing. It appears that there is XPD recruitment even when GTF2H1 is expressed. The authors should provide quantitation as well as a western demonstrating comparable expression of rescue in the cells used.

We have now included a quantification of the experiments shown in Fig. 4a,b, which is presented in the new Fig. 4b and clearly shows significant rescue of XPD recruitment in cells expressing GTF2H1. The reviewer writes '*It appears that there is XPD recruitment even when GTF2H1 is expressed*'. This remark is unclear to us. It is obvious that there is XPD recruitment when GTF2H1 is expressed, as this is the normal situation. Possibly, the reviewer meant '*when GTF2H1 is not expressed*'. In response to this, we would like to point out that we do not claim that there is absolutely no XPD recruitment when GTF2H1 levels are lowered by BRM or BRG1 depletion. We only show that the XPD recruitment is reduced, as

is also shown in the quantification in Fig. 1f. This is similar as what is observed in the experiments shown in Fig. 4.

It is also unclear to us what the reviewer means with a western ‘*demonstrating comparable expression of rescue in the cells used*’. These experiments show an siRNA-treated mixed population of cells with or without transiently expressed GFP-GTF2H1 (or XPB-GFP). The essence and strength of this ‘single cell’ experiment is that it allows us to directly compare (and quantify) signals in individual cells (i.e. accumulation at local damage) to those in neighboring cells within the same image field. In other words, in the same IF experiment we can compare XPD accumulation in cells that express GFP-GTF2H1 (or XPB-GFP) to that in cells that do not express GFP-GTF2H1 (or XPB-GFP). We thus cannot use such a mixed cell population to investigate protein expression levels by western blotting.

7. Fig. 5 – it is claimed that BRG1 knockdown only lowers GTFH1 in U2OS but not A549 and H1299 cells; however, due to the low baseline GTFH1 levels, this claim is insufficiently supported. The authors should validate knockdown in these cells in another manner perhaps by qRT-PCR.

The original figure to which the reviewer is referring is shown below. It is unclear to us why the reviewer calls the GTF2H1 levels ‘*low baseline*’, as clear GTF2H1 staining is observed (similar to U2OS). GTF2H1 is not reduced after siBRG1 in A549 and H1299 but is clearly reduced after siBRM. However, to further confirm this finding, we have repeated this experiment and replaced this figure, in the revised manuscript, and added quantification from multiple independent experiments (Fig. 5a,b, respectively), again clearly showing that GTF2H1 levels in A549 and H1299 cells are not reduced after siBRG1. This is also not surprising, as these cells lack BRG1, which is also described in literature (Strobeck et al, J Biol Chem, 2002; Medina et al, Hum Mutat, 2008; Hoffman, PNAS, 2014). As we clearly do not observe any effect of siBRG1 in these BRG1-deficient cells, we do not think that performing the proposed qRT-PCR experiments is necessary or will provide any additional relevant insight.

8. Fig. S4B – CRISPR/Cas9 does not appear to be knocking out BHM expression. The authors claim that there is decreased levels in H1299 but this not supported by their data. The authors should validate genetic knockout in all alleles.

We apologize for the confusion that may have arisen as we may not have clearly described how this experiment was set up. The cells shown in the original western blot of

Supplementary Fig. 4b were a heterogeneous population of cells transduced with sgBRM (or sgBRG1 or control) and thus showed reduced BRM expression (clearly visible in the western blot after sgBRM) over the entire population. As this was a mixed population of transduced cells, some cells may have full knockout and all alleles targeted while other cells in the population may only have knockdown because not all alleles are mutated. As this was a mixed population, it would also have been useless to validate '*genetic knockout in all alleles*'. To avoid the confusion caused by using this heterogeneous population, we have now repeated these experiments studying the impact of BRM depletion on survival of BRG1-deficient A549 and H1299 cells using BRM and BRG1 siRNAs instead of sgRNAs. These new experiments replace the original experiments in the revised manuscript and are shown in Fig. 4c-e and Supplementary Fig. 4c-e. A clear reduction of BRM in A549 and H1299 after siBRM is shown on western blot. Similar to the original experiments with sgRNAs, we again find that depletion of BRM severely impairs proliferation of BRG1-deficient A549 and H1299 cells, but this is not rescued by ectopic expression of GTF2H1.

9. Fig. 5 – levels of GTFH1 should be shown by western blot analysis rather than IF which is less quantitative and more subjective to interpretation of expression levels.

As Fig. 5 does not show any IF, we assume that the reviewer means either (original) Fig. 6c or Supplementary Fig. 5 that depict IFs of GTF2H1 levels. In the original manuscript, we showed the same GTF2H1 levels in MRC5 sgBRM clones c1 and c3 both by IF (Fig. 6c, quantified in Fig. 6d to avoid 'subjective interpretation') and western blot (Supplementary Fig. 5c). This clearly indicates that both techniques are equally quantitative. Moreover, in the revised manuscript, we have repeated the western blot, in which we now also show GTF2H1 levels in the two other sgBRM clones c6 and c7. These repeated experiments, with quantifications, again indicate that quantified IFs display the same difference in GTF2H1 levels as western blot (Fig. 6c-f in the revised manuscript). Thus, there is no clear rationale to assume that IF is less quantitative or more subjective. On the contrary, IF is much more informative as it reveals cell to cell variation. This is important, because we find that the reduced levels of GTF2H1 after chronic BRM depletion are either retained or rescued in individual cells. Visualization of this cell to cell variation is certainly not possible with western blot analysis. However, as requested and to confirm our IF observations, in the revised manuscript, we now show for all the clones that we functionally test for DNA damage sensitivity (c1, c3, c6 and c7), by western blot the same differences in GTF2H1 levels as shown by quantified IF.

10. Authors should expand on their key finding that BRM/BRG1 regulates transcription of GTF2H1. At minimum, CHIP-Seq experiments should be performed to validate that SWI/SNF is acting specifically at this locus in response to UV. (Fig. 3)

We thank the reviewer for this suggestion. As ChIP-seq experiments with BRG1 and BRM have been performed by multiple groups before us, we re-analyzed ChIP-seq data from two publicly available datasets (Raab et al, Epigenetics Chromatin, 2017; Raab et al, bioRxiv 322065, 2018) to show that both BRM and BRG1 are present at the GTF2H1 locus and enriched at its promoter. In the revised manuscript, this is shown in Fig. 3c and Supplementary Fig. 3c. It is, however, important to stress that we nowhere show nor claim that SWI/SNF complexes act at the GTF2H1 locus in response to UV specifically, as the reviewer incorrectly suggests. Our data indicate that BRM and BRG1 act to maintain GTF2H1 expression levels in unperturbed cells. In the absence of BRM or BRG1, GTF2H1 levels are diminished, which leads to unstable TFIIH and, as a consequence, impaired NER and hypersensitivity to UV irradiation. We do not claim BRM or BRG1 act specifically in response to UV to regulate GTF2H1.

11. Is there a correlation between loss of GTF2H1 and BRM/BRG1 deficiency in cancer

patients? And if so, does this deficiency translate to cisplatin sensitivity? Mining data from the cancer therapeutics response portal would be useful to address this.

Based on our data, it seems an excellent idea to figure out whether reduced GTF2H1 expression is observed in BRM/BRG1 deficient cancer and to correlate this to platinum-drug sensitivity, as we have also indicated at the end of our discussion and in our model (Fig. 7). However, we think that such an in-depth, more clinically oriented study is beyond the scope of our manuscript, which primarily addresses the question how BRM and BRG1 loss impacts NER efficiency. We did check the cancer therapeutics response portal and thank the reviewer for this thoughtful suggestion. However, we did not find any data useful to indicate whether a correlation exists or not between GTF2H1 levels and cisplatin or carboplatin sensitivity.

12. Fig 5c-e: dose response curves should be shown, not just data from one dose.

It is unclear to us what the reviewer means and we apologize for any confusion that may have arisen if we have not been clear. The experiments depicted in the original Fig. 5c-e were performed in the absence of any 'dose' or DNA damage. In these experiments, cells were only 'treated' with sgRNAs, either control or against BRG1 or BRM. These experiments were performed to test whether synthetic lethality of BRG1-deficient cells after loss of BRM (as observed in literature and by us) could be explained by reduced GTF2H1 levels. These experiments therefore do not involve nor need any 'dose' or DNA damage. As explained above, we have replaced these experiments by new experiments using siRNA instead of sgRNA, leading to the same results.

Minor Comments:

1. Fig 2B – needs nucleoplasmic marker

These images are stills from a time-lapse imaging series of living cells expressing XPB-GFP, as clearly indicated in the legends. As such, a nucleoplasmic marker such as DAPI is not applicable. In addition, XPB (endogenous or tagged with GFP) is well known to be expressed exclusively in the nucleus (Hoogstraten, 2002, Mol Cell; Giglia-Mari, 2009, PloS Biology). As such, XPB is by itself already a nucleoplasmic marker. We have indicated the known exclusive nuclear expression of XPB clearly in the legend.

2. There is reference to a Fig. 4C in the figure legend but no corresponding pic or reference in the Results section.

We thank the reviewer for noticing this mistake and have corrected it.

3. There is incorrect referencing of figures – i.e. “Depletion of BRG1 led to lower overall transcription (Fig. 3E)”

We have carefully checked all our figure references in the revised manuscript. As for the indicated incorrect reference, we do not think that this reference is incorrect. Fig. 3e, which is now Fig. 3f in the revised manuscript, clearly shows lower EU incorporation, i.e. overall transcription levels, after siBRG1. Possibly confusion has arisen because transcription is indicated by 'relative EU incorporation' in the figure. We have indicated this now more clearly in the legends.

4. Most figures contain error bars, but importantly, lack statistical analysis: see Fig. 1a,c; 2a,c,e; 3b,c,e; 5d,e; 6b,h,i. Moreover, a number of studies were only completed with 2 replicas.

As discussed above, and in response to the other reviewers, we have repeated several key experiments (for instance, experiments now shown in Fig. 3a, 3d, 5a, 5c, 5d, 6e, Supplementary Fig. 4c, 4e). In addition, we performed and indicated statistical analysis for each experiment if applicable (i.e with at least 3 data points).

5. Fig S5c : Western data for sgBRM clones c6 and c7 should be shown.

We show the western blot for clone c6 and c7 in Fig. 6e of the revised manuscript.

Reviewer #2 (Remarks to the Author):

This paper by Ribeiro-Silva and colleagues elaborates on the impact of SWI/SNF ATPases on DNA damage sensitivity. In their study the authors demonstrate that BRM and BRG1 facilitate the expression of GTF2H1. This functional relationship is examined in selected cancer cell lines where the authors observe that, despite the loss of the ATPases, GTF2H1 levels are restored. The authors speculate that this might open an avenue for therapeutic intervention.

The functional relationship between the SWI/SNF and GTF2H1 is interesting but also quite trivial. SWI/SNF complexes have a broad spectrum of target genes and it is not surprising that DNA repair factors are amongst them. What about other SWI/SNF target genes that might cause the observed phenotypes in cancer cells? Just to give an example, I believe a genome-wide view would have made a stronger point than selecting NER target genes for qPCR analysis.

We do agree that SWI/SNF complexes will regulate a host of genes, but we do not understand why our study is trivial. The reviewer may have overlooked in the introduction the rationale of our study, where we refer to our previous work showing that SWI/SNF is important for the UV-induced DNA damage response (Lans et al, PLoS Genetics, 2010). Since NER is the sole DNA repair pathway in mammals that is able to remove UV photolesions (Marteiijn et al, Nature Reviews MCB, 2014), it is utterly logical to focus on NER. In addition, we made a big effort (Fig. 1-4) on describing and showing how we came to identify GTF2H1 as target of BRM, based on the observed phenotypes. This is therefore not simply 'selecting NER target genes for qPCR analysis', as other target genes would not readily explain the observed phenotypes. We certainly do not claim that BRM and BRG1 do not have many more target genes, as this was indeed shown by multiple genome-wide studies before us. However, it would have been 'trivial' for us to merely repeat these previous genome wide analyses. Importantly, these previous genome-wide analyses, albeit GTF2H1 was among the target genes of BRM and BRG1 (see for instance Raab et al, Epigenetics Chromatin, 2017; Raab et al, bioRxiv 322065, 2018), have not followed up on or revealed any functional relationship between BRM/BRG1 and GTF2H1, as we do in our manuscript. GTF2H1 is a well-known stabilizing subunit of TFIIH and essential for NER and UV survival (Luo et al, Mol Cell, 2015 and Compe and Egly, Annu Rev Biochem, 2016). We show that ectopic expression of GTF2H1 rescues the NER defect (i.e. TFIIH recruitment to DNA damage) in siBRM and siBRG1 depleted cells and that NER functionality (i.e. TFIIH recruitment to UV-lesions, UV and cisplatin sensitivity) of BRM knockout cells, as is expected based on our results, correlates with the level of GTF2H1 expression. These experiments clearly implicate GTF2H1 as causative factor for impaired NER activity and DNA damage hypersensitivity of BRM or BRG1 deficient cells.

There is no clear indication of a molecular mechanism, which in my opinion would be mandatory to publish this paper in Nature Communications. Does BRM/BRG1 localize to the GTF2H1 locus?

We agree that it is relevant to investigate whether BRM and BRG1 are at the GTF2H1 locus. To that aim, we have re-analyzed ChIP-seq data from two previous genome-wide ChIP-seq analyses for BRM and BRG1 (Raab et al, Epigenetics Chromatin, 2017; Raab et al, bioRxiv 322065, 2018). In the revised manuscript, we now show in the new Fig. 3c and Supplementary Fig. 3c that BRM and BRG1 are both clearly localized to the promoter area of GTF2H1. Furthermore, in our opinion, our manuscript provides a very plausible molecular mechanism for how BRM or BRG1 deficiency leads to NER dysfunction and hypersensitivity to UV irradiation and cisplatin, i.e. through disruption of GTF2H1 expression which limits TFIIH stability and function.

Does BRM/BRG1 have a general impact on the chromatin conformation (ATAC-seq)?

Yes, both BRM and BRG1, via their respective SWI/SNF ATP-dependent chromatin remodeling complexes, are well known to regulate chromatin conformation. SWI/SNF complexes regulate transcription of target genes through (re-)positioning of nucleosomes or maintaining open chromatin to prevent or facilitate binding of transcription factors. This has been shown by others before *in vitro* as well as *in vivo* using different genome wide nucleosome profiling techniques including ATAC-seq (see for instance Tolstorukov et al, PNAS, 2013; Orvis et al, Cancer Research, 2014; Bao et al, Genome Biology, 2015; Zhou et al, Annu Rev Biophys, 2016; Rafati et al, PLoS Biology, 2011; Rawal et al, Genes & Development, 2018).

Eventually, any therapeutic intervention demands not only detailed insight into mechanism but also needs to address potential off-target effects. How do cells respond to the depletion of the ATPases or GTF2H1 ? What about their chromatin conformation? How is the general transcription efficacy affected ?

General transcription will obviously be affected by reduction of GTF2H1 levels, since GTF2H1 is a vital component of the essential transcription initiation factor TFIIH. This is also clearly shown in the original Fig. 3e (now Fig. 3f), which the reviewer has apparently overlooked. Thus, GTF2H1 targeting for therapeutic purposes does not seem to be a good idea, as its depletion will kill any cell, including non-tumor cells. We would like to stress, however, that our manuscript is not about a new therapeutic intervention but about how BRM and BRG1 impact the UV-induced DNA damage response. Importantly, we also do not put forward the idea that BRM, BRG1 or GTF2H1 should be 'depleted' as part of any therapeutic intervention. Thus, any evaluation of off-target effects by their depletion is irrelevant for our manuscript. We apologize if it might have seemed that this idea was promoted, possibly due to our poor choice of words at the end of the abstract (see below; we have adjusted this last sentence). Rather, we express the idea, as a perspective in the discussion, that GTF2H1 could be considered (in future studies) as prognostic marker in cancers with loss of BRM or BRG1, since reduced GTF2H1 renders cells more sensitive to DNA damaging chemotherapeutic agents. Also, at the end of our discussion section, we put forward the idea that the (as of yet unidentified) mechanism by which GTF2H1 expression is restored in chronic BRM/BRG1 deficient cells could be considered as therapeutic target for future therapies. Obviously, these ideas call for in-depth follow up research based on our findings, but are far beyond the scope of our manuscript and were only put forward as part of the end of the discussion.

Similarly, I find the examination of cancer-related phenotypes rather preliminary and not convincing. A pure analysis of protein expression levels and one colony formation experiment cannot make a strong claim. I realize that a lot of work has gone into this research project but I feel that the paper in its current form is rather descriptive and reveals correlations rather than demonstrating clear-cut and convincing data. For a high impact journal like Nature Communications I expect a more detailed analysis and, equally important, a novel and intriguing mechanism.

We are unsure to which 'cancer-related phenotypes' and to which 'claim' the reviewer is referring to. Our manuscript is not a 'pure analysis of protein expression levels' and 'one colony formation experiment'. We thoroughly analyze NER dysfunction, by multiple means, in Fig. 1-4, clearly demonstrating that loss of BRM/BRG1 leads to NER dysfunction because GTF2H1 levels are reduced. We have not made any 'correlations' in these experiments, which are detailed and performed according to accepted standards in the field and which reveal a novel mechanism by which BRM and BRG1 impact NER efficiency. Based on this thorough analysis and the well-known and undisputed fact that GTF2H1 (being an integral component of TFIIH) is essential to NER and thus to cellular UV and cisplatin survival, we

show using multiple and independent colony formation experiments that BRM knockout cells are only sensitive to UV and cisplatin if they have low GTF2H1 levels.

Minor comments

1. In Figure 1 and Figure 2B the knockdown levels of the protein have neither been analyzed by Western blotting nor microscopically.

For all our experiments, we standardly analyze efficiency of protein knockdown by siRNA and the multiple western blots and IFs shown in the manuscript clearly demonstrate that our used siRNAs work efficiently. In the revised manuscript, we have now included the original western blot and IF analyses for Fig 1 and Fig. 2b depicting protein knockdown for the used siRNAs (Supplementary Fig. 1a-c,f).

2. In Figure 2D the XPD and XPB levels do not increase at chromatin but the levels diminish in the nucleoplasm. Figure 3D shows that knockdown of BRM1 has no impact on the global XPD and XPB levels. How can diminished levels of the proteins in the nucleoplasm be explained?

Fig. 3d shows global XPB and XPD levels in cells without UV irradiation. In the western blot shown in Fig. 2d there is only some reduction in XPB and XPD visible in the nucleoplasm after UV irradiation, but there is also some increase in the chromatin fraction after UV. Apparently in this experiment, there is still a small fraction of functional TFIIH that is recruited to damaged DNA. This is not against expectation, but in line with the reduced (not abolished) TFIIH accumulation shown in Fig. 1 and explained by the reduced (not abolished) GTF2H1 levels due to BRM depletion.

3. The quality of many western blots is below standard.

This is a rather bold statement, without being clear which exact western blots the reviewer thinks are below standard. The quality of a number of the presented western blots appeared to be perfectly suited to allow reliable quantifications. However, in the revised manuscript we have repeated several of our experiments and now provide new (even better quality) western blots in Fig. 3d, 5a,c, 6e and Supplementary Fig. 4c.

4. The qPCR analysis on TFIIH subunits is not sufficient. Does BRM act directly on these genes (ChIP experiments) or are these indirect effects?

We agree that the qPCR analysis does not reveal whether BRM acts directly on GTF2H1 or via an indirect mechanism. To address the question whether BRM (and BRG1) can act directly on GTF2H1 to regulate its transcription, we have re-analyzed ChIP-seq data from two previous genome-wide studies. In the revised manuscript, we now show in Fig. 3c and Supplementary Fig. 3c that indeed BRM and BRG1 are localized to the GTF2H1 locus. We also describe these results in the results and discussion sections of the revised manuscript.

5. Figure 4 does not have any quantification nor statistical analysis.

We have quantified multiple replicate experiments as shown in Fig. 4 and have added this quantification with statistics as new Fig. 4b in the revised manuscript.

6. It is stated that BRG1 and BRM may compensate for each others functions. This should be tested.

We indeed did not test whether BRM and BRG1 compensate for each other in regulating GTF2H1. This idea of compensation was based on literature in which such compensation with regard to other functions of BRG1/BRM was described (for instance, Strobeck et al, J. Biol. Chem, 2002; Hoffman et al, PNAS, 2014; Oike et al, Cancer Research, 2013). In the revised manuscript, we have tested whether overexpression of BRG1 can compensate for the loss of BRM, which is now shown in Fig. 6g and Supplementary Fig. 5d. These experiments indicate that with regard to GTF2H1 regulation, the ATPases do not compensate for each other. We have described this clearly in the revised manuscript.

7. The phenotype when assessing the CRISPR mediated knockdown of BRM and BRG1 is not convincing. I would suggest to conduct a more detailed analysis.

In response to this and an earlier concern of the reviewer, we have asked for an explanation of what the reviewer exactly means with 'the phenotype'. This explanation and our response is outlined below.

*In their study the authors present 6 figures and one quite speculative model figure (figure7). Figures 1-4 are technically sound but only describe a fairly simple correlation between BRM and GTF2H1. I would like to refer to the abstract of the manuscript where the authors emphasize the importance of mutations of SWI/SNF genes in human cancers and conclude that their findings "**expose GTF2H1 as a potential novel predictive marker of platinum drug sensitivity and as a therapeutic target in the treatment of SWI/SNF deficient cancers**". I don't think that figures 5 and 6 do justice to this strong statement. I expect more conclusive experiments, regarding the function of BRM in cancers and regarding the treatment of SWI/SNF deficient cancers. At this stage I cannot suggest any particular experiments as I don't know whether they might work out. The current data set (figures 5 and 6 and the respective supplementary figures) is, in my opinion, too preliminary to propose specific experiments.*

We thank the reviewer for his/her willingness to explain his/her concerns in more detail. We agree with the reviewer and certainly did not intend to strongly claim that our findings of Fig. 5 and 6 reveal that GTF2H1 is a therapeutic target in the treatment of SWI/SNF cancers. This sentence was merely meant as a reflection of (the end of) our discussion section, providing a perspective of the potential implications of our results (which is why we used the word 'potential'). However, to avoid any future confusion due to our apparent poor choice of words, we have adjusted this sentence by removing the latter part.

With regard to 'more conclusive experiments regarding the function of BRM in cancers and regarding the treatment of SWI/SNF deficient cancers', as outlined above in more detail, we think that these experiments are beyond the scope of this manuscript. The topic of our manuscript is not '*the function of BRM in cancers*' or '*treatment of SWI/SNF cancers*'. As the reviewer also points out ('*I cannot suggest any particular experiments as I don't know whether they might work out*'), such experiments will not be straightforward and these represent in our opinion an entire new project requiring a different approach and set of techniques. For instance, investigating whether GTF2H1 can be used as diagnostic marker would require access to and systematic screening of primary tumor material derived from platinum chemotherapy treated patients, which is beyond this manuscript.

With respect to the manuscript and the current data, it is not clear to me why the authors use A549/H1299 cell lines to investigate the correlation between BRM and GTF2H1 ? Why only use lung carcinoma cells in the initial examination ? Most papers would probably show a panel of different cell lines or even patient samples to make the claim that GTF2H1 might work as a therapeutic target in the treatment of SWI/SNF deficient cancers. The authors also only work with BRG1 deficient cancer cell lines. What about BRM deficient cancer cell lines ?

The paper focuses on BRM rather than BRG1 so the reader might want to know about these cancers as well.

Again, we would like to point out that we do not claim that GTF2H1 will ‘work as a therapeutic target’ in the treatment of SWI/SNF cancers. We used the A549 and H1299 cell lines as these two cell lines are commonly used to study BRG1 loss and because there is clear documented expression data of BRG1 (and BRM) in these cells (Reisman, Cancer Research, 2003 and Hoffman, PNAS, 2014). In answer to the reviewer’s concern, we have now included an analysis of GTF2H1 levels in a panel of BRG1 and/or BRM deficient cancer cell lines, derived from several types of tumors. This analysis is depicted in Supplementary Fig. 4a,b and suggests that, similar as for A549 and H1299 cells, GTF2H1 levels are not reduced in the BRG1/BRM deficient cancer cells analyzed thus far. Again, analyzing patient samples is at this stage far beyond the scope of this manuscript.

When assessing the lung cancer cell lines the authors state that BRM and BRG1 might compensate for each others function (page 11). This might be a possibility, but I don't see sufficient proof for this hypothesis. Further, in the same paragraph the authors compare lung cancer cells to bone osteosarcoma cells (U2OS), which to me is not comprehensive and certainly not suggestive of an adaptive response.

In answer to this concern, we refer to our response to the same concern above. We merely put forward this idea that BRG1 and BRM might compensate for each other to offer a possible explanation for our findings. As described in the original manuscript, this idea was based on previous literature and not on our own proof. We now included new experiments in the revised manuscript disproving this idea with regard to regulation of GTF2H1 transcription (Fig. 6g and Supplementary Fig. 5d).

As for the comparison of U2OS cells to the A549 and H1299, the new analysis of GTF2H1 levels in multiple cancer cell lines further strengthens the idea that cells with chronic BRG1 and/or BRM loss do not show reduced GTF2H1 levels.

In figures 5b and S4a the authors perform siRNA mediated knockdown of BRM while stably expressing GFP tagged GTF2H1. While the knockdown worked, the reduction of endogenous GTF2H1 is, if at all, very subtle. The authors state (page 11) "siRNA mediated BRM knockdown in these cells only reduced the expression of endogenous GTF2H1". I don't see this. Apart from this, in these figures and also in other figures I expect a quantification and statistical analysis. Also, I don't understand why the authors put this data in the main figures when they later repeat a similar experiment with CRISPR/CAS9 mediated knockout? The siRNA treated cells are not examined in any other way in the manuscript.

We are surprised by this remark as the western blots depicted clearly show reduction of endogenous GTF2H1 (we consistently show reduced GTF2H1 levels, not a complete absence of GTF2H1, which we also never claimed). However, we have repeated these experiments and depict new western blots (Fig. 5c and Supplementary Fig. 4c), which again shown that indeed endogenous GTF2H1 levels are reduced but levels of ectopically expressed GFP-GTF2H1 are not reduced after siBRM. We have now also included a quantification of these experiments (Supplementary Fig. 4d).

Furthermore, we agree that it was not clear why these experiments were performed with siRNA while the subsequent proliferation assays with A549 and H1299 cells were performed after CRISPR/Cas9 knockout. We have repeated these experiments and have investigated the proliferation of A549 and H1299 cells using siRNA rather than sgRNA. These experiments are now shown in Fig. 5d,e and Supplementary Fig. 4e and replace the original sgRNA experiments. This way, the connection between the western blots and subsequent

proliferation assays makes more sense. Importantly, similar to the original proliferation experiments, the new siRNA proliferation experiments confirm that depletion of BRM leads to synthetic lethality in BRG1 deficient cancer cells, which is not rescued by ectopic GTF2H1 expression.

Concerning the CRISPR mediated knockout (figure S4b), I do not see a BRM knockout in A549 cells. The tubulin levels in control cells are higher than in the knockout cells and I see a clear BRM band (lane 3 of figure S4B). Also, the quality of the blot is not acceptable. One cannot decide whether there is a band in the last lane of the BRM blot (H1299 cells). This brings me to an essential question. Did the authors check whether all alleles of the BRM gene have been knocked out when applying CRISPR/Cas9? This is very important as the authors want to convey the message that cancer cells adapt to the loss of one ATPase subunit. Is it adaption or an up-regulation of an allele that was not targeted by CRISPR? Figure S4b (lane 3, BRM blot) would certainly suggest that not all alleles were targeted.

We thank the reviewer for pointing out that we did not sufficiently explain the setup of this experiment. Indeed, there was still some BRM visible in Supplementary Fig.4b (which we referred to as 'efficient overall BRM depletion'), but this was also to be expected as the analyzed cells were a heterogeneous pool of cells transduced with sgRNAs against BRM. As such, not all cells are expected to have full BRM knockout and a residual band on western blot is as expected. As we used these experiments to confirm that loss of BRM in BRG1-deficient cells leads to synthetic lethality, it would thus have been very difficult (if not impossible) to establish clonal cell lines with full BRM knockout from this mixed population, as these are expected to be lethal. Because the western blot analyzed a heterogeneous mixture of cells (with likely different indel mutations in each cell), it would have been pointless to check whether all alleles of the BRM gene had been knocked out. However, as explained above, we have replaced these experiments with proliferation assays performed after siRNA-mediated knockdown. Thus, as the original Supplementary Fig. 4b is no longer part of the revised manuscript, we think that this concern of the reviewer is no longer applicable and we expect that the clarity of the setup of these proliferation experiments is now better explained.

Further, in figure S4b there is no examination of the GFP-GTF2H1 levels or endogenous GTF2H levels. Hence, this experiment is not sufficiently controlled and one cannot assess the outcome of the experiment shown in figures 5c-e. Further, the effect of re-expressing GTF2H1 in A549 cells (fig 5d) is quite subtle. Again, also having the aforementioned in mind, this must not necessarily be an indication of an adaption. The statement that (page 12) "these results indicate a synthetic lethality induced by BRM depletion in BRG1-deficient cancer cells is not solely dependent on GTF2H1 expression" cannot be drawn for the above mentioned reasons. But even if so, I would like to know what else it might depend on.

As explained above, we have replaced the original experiments of Supplementary Fig. 4b and Fig. 5c-e. In the new experiments, the levels of GFP-GTF2H1 after siRNA were determined and quantified (Fig. 5c, Supplementary Fig. 4c,d) and these experiments are therefore now better controlled. As for the effect of re-expressing GTF2H1 in A549 cells, we completely agree that this effect was subtle (as also indicated by us in the original manuscript with 'only partially rescued') but we never claimed or stated that this was 'necessarily an indication of an adaptation'. This subtle effect is also not observed in the new experiments, which again show that the synthetic lethality induced by BRM depletion in BRG1-deficient cancer cells is not rescued by ectopic GTF2H1 expression. We refer to the experiments shown in Fig. 6 and Supplementary Fig. 5 as clear indication that cells that chronically lack BRM can adapt and restore GTF2H1 levels.

Synthetic lethality between BRM and BRG1 is not exclusively shown by us (see for instance Strobeck et al, J Biol Chem, 2002; Hoffman, PNAS, 2014) and is also not the main topic of

our paper. We do not know which precise function of BRG1 and BRM is responsible for this synthetic lethality, but as we show it is not via regulation of GTF2H1, which is the topic of our paper, we do not think it is useful to investigate this further as part of this manuscript.

Concerning the different clones assessed in figures 6 and S5. Despite the clear knockout in the western blot (figure 5C), it is not clear whether all BRM alleles have been targeted by CRISPR/Cas9 in these clones. Should this be the case, I would be interested to know how these cells recover GTF2H1 levels or, in other words, I would like to see an indication of a molecular mechanism for the adaptation.

We have sequenced the clones that we functionally analyzed (c1, c3, c6 and c7) to show that all alleles of BRM in these cells were targeted (now shown in Supplementary Fig. 5c). Importantly, all identified mutations are predicted to lead to early truncation of BRM (deleting all known BRM domains), which we confirmed by repeating western blots for these clones clearly showing the absence of BRM protein. Our analysis of what happens to GTF2H1 levels in cells with chronic BRM knockout provides a clear indication that cells apparently can recover GTF2H1 levels. This provides a plausible explanation why the tested cancer cells do not show reduced GTF2H1 levels. Obviously, it would be very interesting to know how these cells then recover GTF2H1 levels, but to investigate which mechanism or mechanisms (as there could be multiple) are responsible constitutes an entire new research project that is beyond the topic and scope of the current manuscript, which deals mainly with how BRM and BRG1 deficiency impairs NER. This would require either the systematic characterization of all different transcriptional regulators present at the GTF2H1 locus in BRM or BRG1 deficient cells or, even better, an unbiased (whole genome) screen aimed at identifying the factors responsible. We have already tested some potential candidates, such as for instance EZH2 (as also described in the discussion), but this has thus far been without success.

Coming back to my previous point. Although the experiments with cisplatin are technically sound, I would like to see more experiments to support the claims made in the abstract and throughout the manuscript.

We are confident that our detailed explanations in response to these concerns, our correction of the last sentence of the abstract, the added new experiments, western blots and quantifications (several also in response to the other reviewers) and adjustments made to the text of manuscript, sufficiently support the claims made throughout our paper.

Reviewer #3 (Remarks to the Author):

SWI/SNF chromatin remodeling complexes are highly mutated in cancers. Great efforts are underway to understand their function and their possible involvement in the carcinogenic process. In their manuscript "DNA damage sensitivity of SWI/SNF-deficient cells depends on TFIIH subunit p62/GTF2H1" Ribeiro-Silva et al tackle this challenge by addressing the role of the two ATPase subunits of SWI/SNF complexes in nucleotide excision repair (NER).

.....

In general, I find this to be an important paper that initially set out to investigate the role of BRM in DNA repair - and found that the role of BRM in repair is primarily a byproduct of its role in transcription. They uncover a mechanism by which mutations in BRM and BRG1 alter transcription levels in cells and as a result, the ability to repair DNA. Understanding the function of SWI/SNF proteins, that are highly mutated in cancer is extremely important both to understanding their contribution to cancer risk and to identify possible cancer-specific targets for therapy.

We thank the reviewer for his/her clear summary of our main findings and pointing out the importance of our findings.

Major comments:

1) Previous reports indicated that BRG1 was involved in repair of CPD and not (6-4)PP - indicating it may be specifically involved in TC-NER and not GG-NER (Zhao Q. et al, JBC 2009, Gong F. et al, Cell Cycle 2008, DOI: 10.4161/cc.7.8.5647). The authors measure repair using the unscheduled DNA synthesis assay that measures total NER across the genome, and see a reduction after knock down of BRM and BRG1. They draw the conclusion that GG-NER is therefore affected. However, for the major UV damage - CPD - repair in early time points is primarily transcription coupled. Therefore, theoretically, even if BRM/BRG1 knock down solely affected TCR would you not see this effect? Could you also knock down CSB and show that in the absence of CSB BRG/BRM knock down also affects repair - showing it really is affecting GG-NER?

Indeed, TC-NER more efficiently removes CPDs than GG-NER. However, TC-NER only occurs in actively transcribed strands of genes which is a very minor part of the whole human genome. This is why in TC-NER deficient cells (such as CS-B cells) UDS is not clearly reduced. For the same reason, TC-NER defects are also not expected to lead to any clear visual reduction in recruitment of downstream NER factors to sites of local UV irradiation, as shown in Fig. 1e,f. Moreover, we show that TFIIH complex function is generally impaired after BRM or BRG1 depletion. This will affect both GG-NER and TC-NER as TFIIH is essential for both processes. To confirm, however, that BRM and BRG1 depletion affect GG-NER, we have conducted a UDS assay in TC-NER deficient CSB patient fibroblasts (CS1AN). This experiment, which is shown on the right, clearly shows that UDS efficiency is still reduced by siBRM and siBRG1 in the absence of TC-NER.

2) The results indicate that BRM primarily affects repair through its stabilizing of TFIIH - and GTF2H1 expression. Since it does not interact directly with TFIIH - one possible mechanism would be by enhancing its transcription (binding the GTF2H1 promoter). CHIP of these subunits may be tricky, but there are of genome-wide ChIP-seq data sets available that could be useful such as doi: 10.1186/s13072-017-0167-8

We thank the reviewer for this constructive suggestion. In the revised manuscript, we have re-analyzed whole genome ChIP-seq data from the suggested study and from another one as well (Raab et al, bioRxiv 322065, 2018). In the revised manuscript, we now show in Fig. 3c and Supplementary Fig. 3c that BRM and BRG1 both occupy the promoter of GTF2H1.

Minor comments:

1) Figure 1: For each siRNA condition, the untreated cells are considered as 100%. Since BRG1 and BRM result in a general reduction of RNA synthesis, would it not be a more accurate representation of the to normalize all samples to untreated siControl?

It is indeed true that in the transcription recovery assay, the overall reduction in RNA synthesis after siBRM and siBRG1 is also apparent. However, this particular assay is aimed at determining whether transcription recovers back to levels before irradiation (no UV) levels. This is why we need to normalize to pre-damage condition for each individual siRNA. This is also accepted practice in the field. See for instance RRS examples from different labs Schwertman, Nature Genetics, 2012; Kashiya, Am J Hum Genet, 2013; Mourgues, PNAS, 2013; Niida, Nature Communications, 2017; Pines, Nature Communications, 2018.

2) Figure 3: First, a suggestion to change the order of the figures - to first show RNA and protein expression is lower, and then XPB stability is lower. Second, I think in Fig3b: I would name the plot XPB stability and not turnover, because turnover was not specifically measured, rather the protein levels.

We thank the reviewer for this suggestion and have changed the order accordingly. In this section, we also included the description of the ChIP-seq analysis. We have also adjusted the title of the plot.

3) I would like to see quantifications of figure 4 since the effects are partial and not complete.

We have now included quantifications of the experiments shown in Figure 4A and 4B. This is shown in Fig. 4b.

4) Figure 6c shows the different levels of GTF2H1 in the two clones assayed. However, it is important to show that indeed in the same cells BRM is indeed knocked out. This data is currently in supplemental figure 5.

We have replaced Fig. 6c with new IF images of the two clones (now also showing the other two clones c6 and c7) showing simultaneous staining for both GTF2H1 and BRM, indicating clear BRM knockout. In addition, we added a new western blot depicting GTF2H1 levels and BRM knockout in the main figures (Fig. 6e).

5) Figure 7: Could the general lower RNA expression in TFIIH deficient cells not further reduce BRM/BRG1 expressio - leading to an additional negative feedback?

The scenario indicated by the reviewer is interesting and certainly a possibility. This may also be exactly the reason why there may be strong (selective) pressure for cells to recover GTF2H1 levels, as this would otherwise lead to even more severe transcription defects. Of course, lowering GTF2H1 may not only further reduce BRM and BRG1 expression, but also the expression of other genes that may or may not be involved in transcription. We tested whether 48 h after siRNA depletion of GTF2H1, there may already be any reduction in BRM or BRG1 expression, but we did not observe this clearly (see below). We think that including this hypothetical scenario in the model would make the model possibly too complex and too speculative. This is why we prefer to keep the model as it is.

REVIEWERS' COMMENTS:

Reviewer #1 (Remarks to the Author):

The authors have done a satisfactory job in addressing my technical concerns; however, I still have some concerns on the overall conceptual advance. Kathandapani, Exp Cell Res, 2012 and Bell, Clinical Cancer Research, 2016 have both shown that BRG1 and BRM are important for determining cisplatin response, which is known to be repaired by NER. The novelty of the authors's findings apparently would be in finding that both of these SWI/SNF ATPase regulate expression of GTF2H1, at TFIIH component, which as reviewer 2 points out may not be that surprising. Is this finding that insightful and impactful? The authors argue that maybe GTF2H1 may be a better biomarker of cisplatin response but wouldn't this hold for any downstream central NER component?

Reviewer #2 (Remarks to the Author):

In this revised version the authors have addressed some of my concerns experimentally, which is much appreciated. The suggestions of the other reviewers have also been considered and hence the manuscript has improved significantly. The experimental design is sound and experiments are now presented with a detailed and comprehensive statistical analysis.

In agreement with reviewer #1 ("the overall conceptual advance is relatively incremental") my impression was that the main story of this paper was not a good fit for Nature Communications. For this reason, I also suggested to include more conclusive experiments regarding the role of BRM in cancer, which would have given this paper an important and interesting spin. The authors state that such experiments were beyond the scope of this paper. This is somewhat unsatisfactory but still a valid point of view.

Given the amount of work that has gone into the revision and the technical soundness of this paper, I think that it is in a state that allows publication. I leave it to the editorial board whether Nature Communications is the appropriate platform to communicate this research.

Reviewer #3 (Remarks to the Author):

I am satisfied with the authors response to my comments and find the paper to be improved by their revisions, and recommend the paper for publication in Nature communications.

We would like to thank all three reviewers for the time they invested in the review process of this manuscript and for their generally constructive criticism and suggestions that have allowed us to improve our manuscript. Please find below our point-to-point response to the final reviewer's comments.

Reviewer #1 (Remarks to the Author):

The authors have done a satisfactory job in addressing my technical concerns; however, I still have some concerns on the overall conceptual advance. Kathandapani, *Exp Cell Res*, 2012 and Bell, *Clinical Cancer Research*, 2016 have both shown that BRG1 and BRM are important for determining cisplatin response, which is known to be repaired by NER. The novelty of the authors's findings apparently would be in finding that both of these SWI/SNF ATPase regulate expression of GTF2H1, at TFIIH component, which as reviewer 2 points out may not be that surprising. Is this finding that insightful and impactful? The authors argue that maybe GTF2H1 may be a better biomarker of cisplatin response but wouldn't this hold for any downstream central NER component?

We are happy to read that we have sufficiently addressed all technical concerns raised. We regret that this reviewer still questions the impact of our findings, by arguing that it may not be that surprising that GTF2H1 gene expression is regulated by SWI/SNF. However, we do understand that our points raised to stress the novelty of our findings are somewhat drowned in the rather lengthy point-by-point rebuttal, because of which we summarize these again. Although SWI/SNF subunits are known to regulate gene expression, it is a novel finding that both BRG1 and BRM regulate the expression of GTF2H1, but not that of other TFIIH or NER factors. It is also novel that BRM, by regulating GTF2H1, is important for nucleotide excision repair function, and thus for the DNA damage response to genotoxic agents like UV and platinum drugs. Previously, only BRG1 was linked to nucleotide excision repair, but its importance to nucleotide excision repair by controlling GTF2H1 was also not known. Furthermore, our work confirms that BRM and BRG1 are important for the cisplatin response, but we demonstrate that cells with chronic deficiency of SWI/SNF ATPases BRM and BRG1 can restore GTF2H1 expression. Because of this, BRM- or BRG1-deficient cells will show increased sensitivity to drugs like cisplatin only if GTF2H1 levels are lower than in wild-type cells. Consequently, the absence of BRM or BRG1 might not always be a reliable indicator of cisplatin sensitivity as suggested in Kathandapani, *Exp Cell Res*, 2012 and Bell, *Clinical Cancer Research*, 2016. Since we observed a specific regulation of GTF2H1 expression by BRM/BRG1 but not of other downstream central NER components, GTF2H1 itself is an evident indicator of the cisplatin response, based on its expression levels alone. Lower GTF2H1 levels act as a limiting factor to the recruitment of downstream NER components, but because these other NER factors are not transcriptionally regulated by BRG1 or BRM, their expression level cannot be used as biomarker (in SWI/SNF deficient cancers).

Reviewer #2 (Remarks to the Author):

In this revised version the authors have addressed some of my concerns experimentally, which is much appreciated. The suggestions of the other reviewers have also been considered and hence the manuscript has improved significantly. The experimental design is sound and experiments are now presented with a detailed and comprehensive statistical analysis.

In agreement with reviewer #1 ("the overall conceptual advance is relatively incremental") my impression was that the main story of this paper was not a good fit for Nature Communications. For this reason, I also suggested to include more conclusive experiments regarding the role of BRM in cancer, which would have given this paper an

important and interesting spin. The authors state that such experiments were beyond the scope of this paper. This is somewhat unsatisfactory but still a valid point of view.

Given the amount of work that has gone into the revision and the technical soundness of this paper, I think that it is in a state that allows publication. I leave it to the editorial board whether Nature Communications is the appropriate platform to communicate this research.

We would like to thank the reviewer for his/her appreciation of our efforts in addressing his/her concerns, as well as those of the other reviewers, which allowed us to improve our manuscript. We also appreciate that the reviewer recognizes our thorough experimental design and detailed analysis in the revised manuscript and considers our point of view valid.

As explained in our reply to Reviewer 1, we stand by the novelty of our findings. We are confident that our evidence, as presented in this manuscript, that the expression of GTF2H1 is controlled by the SWI/SNF ATPases BRM and BRG1 holds promise for future research. Due to the high incidence of SWI/SNF mutations in cancers, the indirect effect of BRM and BRG1 on Nucleotide Excision Repair (lower repair, increased sensitivity) can be more easily assessed via GTF2H1 levels, which inversely correlate with sensitivity to DNA damage, rather than looking to each of the SWI/SNF ATPases individually. Cells with chronic SWI/SNF inactivation can recover GTF2H1 levels, highlighting an unknown mechanism that can replace SWI/SNF activity in the regulation of GTF2H1 expression. When and how this mechanism is activated and why not in all cells are interesting and promising research questions. Answering these might provide the insights to reverse SWI/SNF-deficient cells that show restored GTF2H1 levels and Nucleotide Excision Response activity back to a hypersensitivity to DNA damaging agents. However, investigating which compensation mechanism or mechanisms (as there could be multiple) is beyond the scope of our paper which deals with the basic biological problem of how BRM- and BRG1-deficiency impairs Nucleotide Excision Repair, as discussed in the previous rebuttal letter.

Nonetheless, we wish to express our gratitude to the reviewer for accepting the differences of perspective and for the high level of intellectual discussion.

Reviewer #3 (Remarks to the Author):

I am satisfied with the authors response to my comments and find the paper to be improved by their revisions, and recommend the paper for publication in Nature communications.

We thank the reviewer for his/hers recommendation to publish the manuscript in Nature Communications.